# Ultrastructural sublaminar-specific diversity of excitatory synaptic boutons in layer 1 of the adult human temporal lobe neocortex

Astrid Rollenhagen[1†], Akram Sadeghi[1†‡], Bernd Walkenfort[2], Claus C Hilgetag[3], Kurt Sätzler[4], Joachim HR Lübke[1,5,6]*

[1]Institute of Neuroscience and Medicine INM-10, Research Centre Jülich GmbH, Jülich, Germany; [2]Medical Research Centre, IMCES Electron Microscopy Unit (EMU), University Hospital Essen, Essen, Germany; [3]University Hospital Hamburg-Eppendorf, Center for Experimental Medicine, Institute for Computational Neuroscience, Hamburg, Germany; [4]School of Biomedical Sciences, University of Ulster, Coleraine, United Kingdom; [5]Department of Psychiatry, Psychotherapy and Psychosomatics, Medical Faculty/RWTH University Hospital Aachen, Aachen, Germany; [6]JARA Translational Brain Medicine, Aachen, Germany

**\*For correspondence:**
j.luebke@fz-juelich.de

[†]These authors contributed equally to this work

**Present address:** [‡]Research Group Anatomy, School for Medicine and Health Science, Carl von Ossietzky Universität Oldenburg, Oldenburg, Germany

## eLife Assessment

This study provides **important** information on the ultrastructural organization of layer 1 of the human neocortex. The quantitative assessment of various synaptic parameters, astrocytic coverage and mitochondrial morphology is based on **convincing** experimental approaches. These data provide new information on the detailed morphology of human neocortical tissue that will be of interest to neuroscientists working on different network functions.

**Abstract** Layer (L)1, beside receiving massive cortico-cortical, commissural and associational projections, is the termination zone of tufted dendrites of pyramidal neurons and the area of $Ca^{2+}$ spike initiation. However, its synaptic organization in humans is not known. Quantitative 3D models of excitatory synaptic boutons (SBs) in L1 of the human temporal lobe neocortex were generated from neocortical biopsy tissue using transmission electron microscopy, 3D-volume reconstructions, and TEM tomography. Particularly, the size of active zones (AZs) and the readily releasable, recycling, and resting pool of synaptic vesicles (SVs) were quantified. The majority of excitatory SBs contained numerous mitochondria comprising ~7% of the total volume, had a large macular, non-perforated AZ (~0.20 μm²) and were predominantly located on dendritic spines. Excitatory SBs had a total pool of ~3500 SVs, a relatively large readily releasable (~4 SVs), recycling (~470 SVs) and resting (~2900 SVs) pool. Astrocytic coverage of excitatory SBs suggests both synaptic crosstalk or removal of spilled glutamate by astrocytic processes at synaptic complexes. The structural composition of SBs in L1 may underlie the function of L1 networks that mediate, integrate, and synchronize contextual and cross-modal information, enabling flexible and state-dependent processing of feedforward sensory inputs from other layers of the cortical column.

## Introduction

The mammalian neocortex is the most complex part of the brain comprising over 75% of the gray matter. One fundamental feature established during development is the formation of distinct layers with different networks of neurons and its organization into vertically oriented slabs, so-called cortical columns (*Marin-Padilla, 1978*; reviewed by *Rockland and DeFelipe, 2018*). This organization is established in an 'inside layers first-outside layers last' fashion (*Luskin and Shatz, 1985a*, *Luskin and Shatz, 1985b*; reviewed by *Cooper, 2008*). The exception, besides layer (L)6, is L1 generated first originating from the marginal zone (*Marín-Padilla, 1998*; reviewed by *Bystron et al., 2008*). Ongoing cortico- and synaptogenesis finally leads to the formation of the adult cortical network with its unique structural and functional properties (reviewed by *Lübke and Feldmeyer, 2007*; *Rockland and DeFelipe, 2018*).

Remarkably, L1 is highly conserved across cortical areas and higher mammalian species, but its importance was long underestimated. However, L1 is now considered the predominant input layer for top-down information, relayed by a rich, dense network of cortico-cortical, cortico-thalamic, commissural, and associational long-range axonal projections. In humans, and thus different from experimental animals, both L1 excitatory Cajal-Retzius (CR) cells (*Anstötz et al., 2014*) and a quite heterogeneous population of inhibitory neurons (see for example *Jiang et al., 2013*; *Boldog et al., 2018*; *Obermayer et al., 2018*; *Kwon et al., 2019*; reviewed by *Huang et al., 2024*) provide signals to the terminal tuft dendrites of pyramidal neurons of the underlying cortical layers. Thus, L1 is a central locus of neocortical associations controlled by distinct types of inhibition and feedforward excitation (*Schuman et al., 2019*; *Hartung and Letzkus, 2021*, reviewed by *Schuman et al., 2021*).

The temporal lobe neocortex (TLN) is situated on the basolateral aspect of the cerebral hemispheres primarily observed in higher primates, including humans. Notably, in humans, the TLN encompasses approximately 20% of the entire cerebral cortex (*Kiernan, 2012*). It is recognized as a highly specialized associative neocortex, characterized by its homotypic granular six-layered organization (*von Economo and Koskinas, 1925*; *Palomero-Gallagher and Zilles, 2019*). The human TLN plays a crucial role in various cognitive functions, including auditory, visual, vestibular, linguistic, and olfactory processing. Additionally, it is intricately connected to diverse sensory and multimodal associational brain regions, such as the limbic system, amygdala, and various subcortical structures (*Insausti, 2013*). Lastly, the human TLN serves as the epicenter and onset site for temporal lobe epilepsy (*Allone et al., 2017*; *Tai et al., 2018*).

One of the most important discoveries in neuroscience, besides the definition of the neuron by *Cajal, 1911*, was the introduction of the term 'Synapse', more than 100 years ago. Since then, these structures have been investigated from different viewpoints, including structural, functional, molecular, and computational studies summarized in meanwhile thousands of original publications, reviews, and numerous textbooks. Yet, the quantitative geometry of the most prevalent synapse in the brain, the cortical synapse, has remained largely unexplored structurally, in both experimental animals (but see *Rollenhagen et al., 2015*; *Rollenhagen et al., 2018*; *Bopp et al., 2017*; *Hsu et al., 2017*; *Prume et al., 2020*) and, even more so, in the context of the human brain (but see *Yakoubi et al., 2019a*; *Yakoubi et al., 2019b*; *Domínguez-Álvaro et al., 2019*; *Domínguez-Álvaro et al., 2021*; *Cano et al., 2021*; *Cano et al., 2023*; *Schmuhl-Giesen et al., 2022*; *Shapson-Coe et al., 2024*). This could be partially attributed to the availability of human brain tissue samples. Nevertheless, detailed and quantitative descriptions of synaptic complexes in L1 of the human brain are to date non-existent.

Here, an investigation into the synaptic organization of L1 in the human TLN was undertaken using high-resolution transmission electron microscopy (TEM), 3D-volume reconstruction, to generate quantitative 3D-models of synaptic boutons (SBs), and TEM tomography. The ultimate objective of this comprehensive project is to elucidate the synaptic organization, layer by layer, of the cortical column in humans.

Such meticulous quantitative structural analyses of SBs and their corresponding target structures are indispensable for comprehending, elucidating, and establishing connections between the structural and functional properties of the diverse signal cascades underlying synaptic transmission, efficacy, strength, and plasticity. Consequently, these efforts contribute significantly to the understanding of the computational properties of brain networks, in particular the synaptic organization of the cortical column.

## Results

### Neural and synaptic composition of L1 in the human TLN

L1 in the adult mammalian neocortex represents a relatively cell sparse layer, that in humans can be subdivided into two sublaminae, L1a and L1b, as revealed in semithin sections (*Figure 1A*) and Golgi-preparations (*Figure 1B, C, F and G*), as well as at the TEM level (*Figure 1H, I*). L1a is more dominated by a dense network of astrocytes of different shape and size, their fine processes (*Figure 2A*), the occurrence of reactive microglia (*Figures 1H, I and 2A*) and terminal tuft dendrites of pyramidal neurons throughout the neuropil of L1 (*Figures 1H, I and 2B*).

Hence, L1a can be described as a sublamina with a predominant proportion of astrocytes, microglia, and their processes, although it also harbors numerous other structural elements such as terminal tuft dendrites and synaptic complexes. These complexes consist of a SB juxtaposed with either a dendritic shaft or spine (*Figure 2A*) directly beneath or interwoven among the fine astrocytic processes within L1 (*Figure 2A*). Lipofuscin granules, indicative of aging, were frequently noted in astrocytes, reactive microglia (*Figure 2A*), and L1 neurons, although their abundance, size, and morphology varied considerably.

Remarkably, and in contrast to rodents, a persistent subpopulation of excitatory CR-cells was found in humans, identifiable by the horizontal bipolar orientation with intralaminar long-range horizontal axonal collaterals with a transcolumnar projection (*Figure 1E–H*; *Retzius, 1893*; *Retzius, 1894*; *Anstötz et al., 2014*; reviewed by *Martínez-Cerdeño and Noctor, 2014*). The majority were located directly underneath the pial surface (*Figure 1F and G*) or intermingled with a heterogeneous population of GABAergic interneurons (*Figure 1E*).

In the human TLN, GABAergic interneurons constitute the predominant class of neurons in L1, exhibiting significant structural and functional heterogeneity (*Figure 1D and E*; see also *Jiang et al., 2013*; *Verhoog et al., 2016*; *Boldog et al., 2018*; *Obermayer et al., 2018*; *Kwon et al., 2019*; *Schuman et al., 2019*, reviewed by *Huang et al., 2024*) which were found throughout both sublaminae and were sometimes organized in clusters of three to six neurons (*Figure 1D*). Besides CR-cells and GABAergic interneurons, also degenerating neurons, identifiable by their shrunken and dark appearance (*Figure 1A, I*), were found in both sublaminae, although their numbers varied substantially between brain tissue samples.

L1b is primarily dominated by dendritic and synaptic profiles due to a massive increase in synaptic complexes (*Figures 1H and 2B*). Furthermore, L1b is infiltrated by apical dendrites of L2, L3, and L5 pyramidal neurons interspersed with smaller dendritic segments representing apical oblique dendrites (*Figures 1I and 2B*). The majority of synaptic complexes in L1 were axo-spinous (~80%, *Figure 2B, D and E*), the remainder were axo-dendritic, some of which but not all were regarded as putative GABAergic terminals (*Figure 2C*; but see *Silver et al., 2003*).

L1b can be separated from L2 by the sudden increase of neuronal cell bodies and thick apical dendrites originating at the upper pole of the somata of pyramidal neurons (*Figures 1A and 2B*). At the L1b and L2 transition, more and more apical and apical oblique dendrites of different caliber were observable.

### Synaptic organization of L1 in the human TLN

The primary objective of this study was to investigate the synaptic organization of excitatory SBs in L1 in the human TLN, focusing on synaptic parameters that represent possible morphological correlates of synaptic transmission, efficacy, strength, and plasticity. The quantitative analysis was separated for both sublaminae to look for possible sublaminar-specific differences. Besides the overall geometry of SBs and the number and size of mitochondria, in particular the size of the PreAZ and PSD constituting the AZ, the morphological equivalent to a functional neurotransmitter release site, and the organization of the three functionally defined pools of SVs, namely the readily releasable (RRP), the recycling (RP), and resting pool, were quantified.

To achieve this, a total of 361 SBs was completely reconstructed out of four patients in six series containing 100–150 ultrathin sections/series. A total of 190 SBs and 184 AZs (L1a) and 171 SBs and 165 AZs (L1b) using neocortical access tissue (see Material and Methods; *Source data 2*) were completely 3D-reconstructed and quantified using fine-scale TEM, digital TEM images, and subsequent 3D-volume reconstructions.

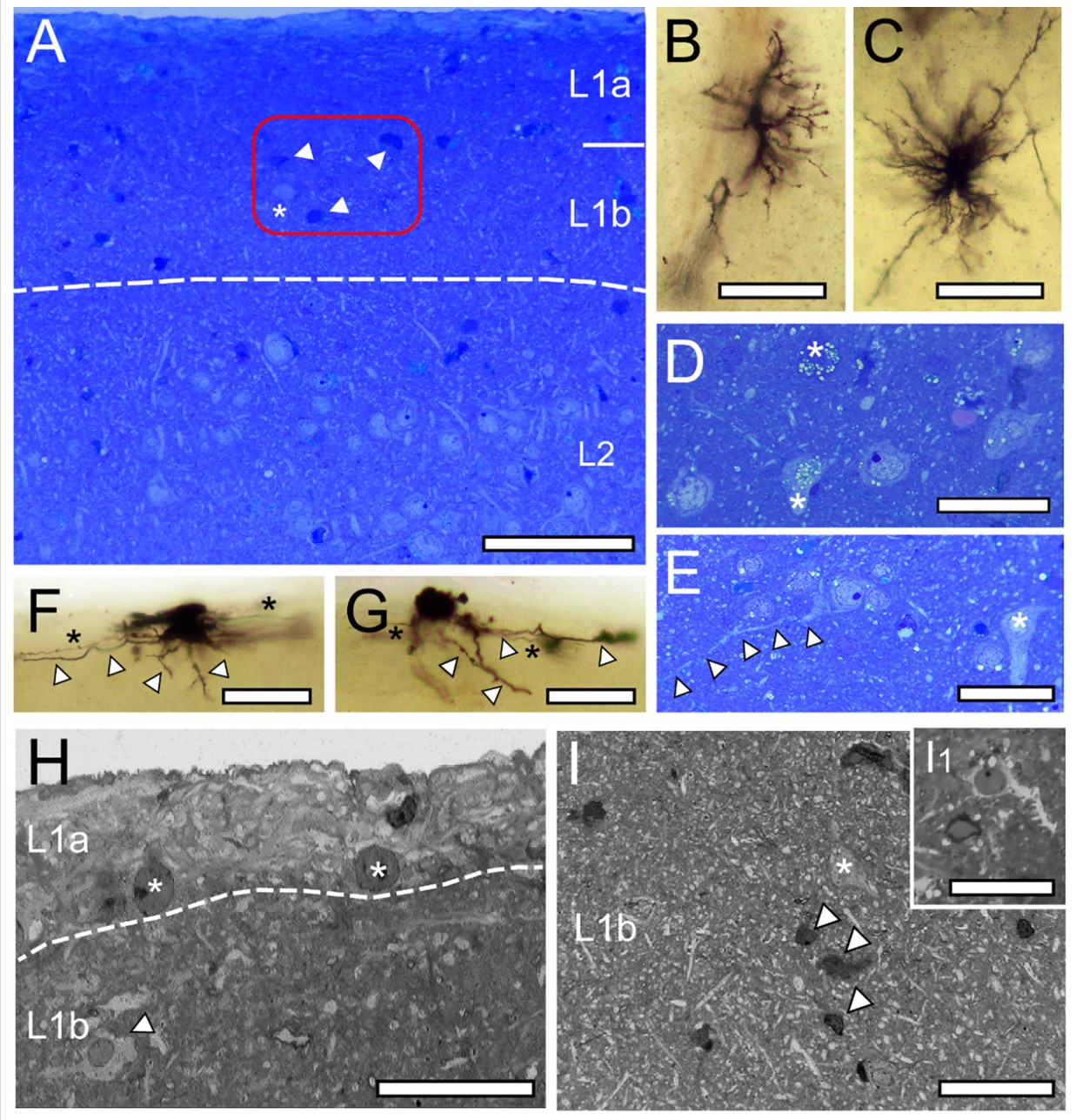

**Figure 1.** L1 and L2 in the human TLN as revealed in methylene-blue stained semithin sections, Golgi-impregnation, and low power TEM. (**A**) Methylene-blue stained semithin section of L1a, L1b, and L2. The cell sparse zone contains degenerating neurons (dark appearance, arrowheads in the framed area) and a cell body (asterisk in the framed area). Note the sudden increase in the density of neurons at the L1b/L2 border. Scale bar 500 µm. (**B, C**) Golgi-impregnated radial astrocytes in L1a (**B**) and L1b (**C**). Scale bars 50 µm. (**D**) Group of neurons in L1b containing lipofuscin granules of various shape and size (asterisks) most prominent in two of the neurons. Scale bar 50 µm. (**E**) Putative CR-cell identifiable by its long horizontally oriented dendrite (marked by arrowheads) and an inverted putative GABAergic interneuron (asterisk) in L1b. Scale bar 50 µm. (**F, G**) Golgi-impregnated CR-cells in L1a with the characteristic horizontal orientation of dendrites (arrowheads) and axons (asterisks). Scale bars 50 µm. (**H**) TEM micrograph of L1a underneath the pial surface containing two putative CR-cells (asterisks). Note the abrupt transition (indicated by the dashed line) to the neuropil of L1b, a reactive microglial cell (arrowhead), and dendrites and synapses. Scale bar 25 µm. (**I**) TEM micrograph of L1b containing thousands of dendritic profiles, a putative GABAergic interneuron (asterisk) but also degenerating neurons (arrowheads) and a reactive microglia ( I 1) Scale bar 50 µm.

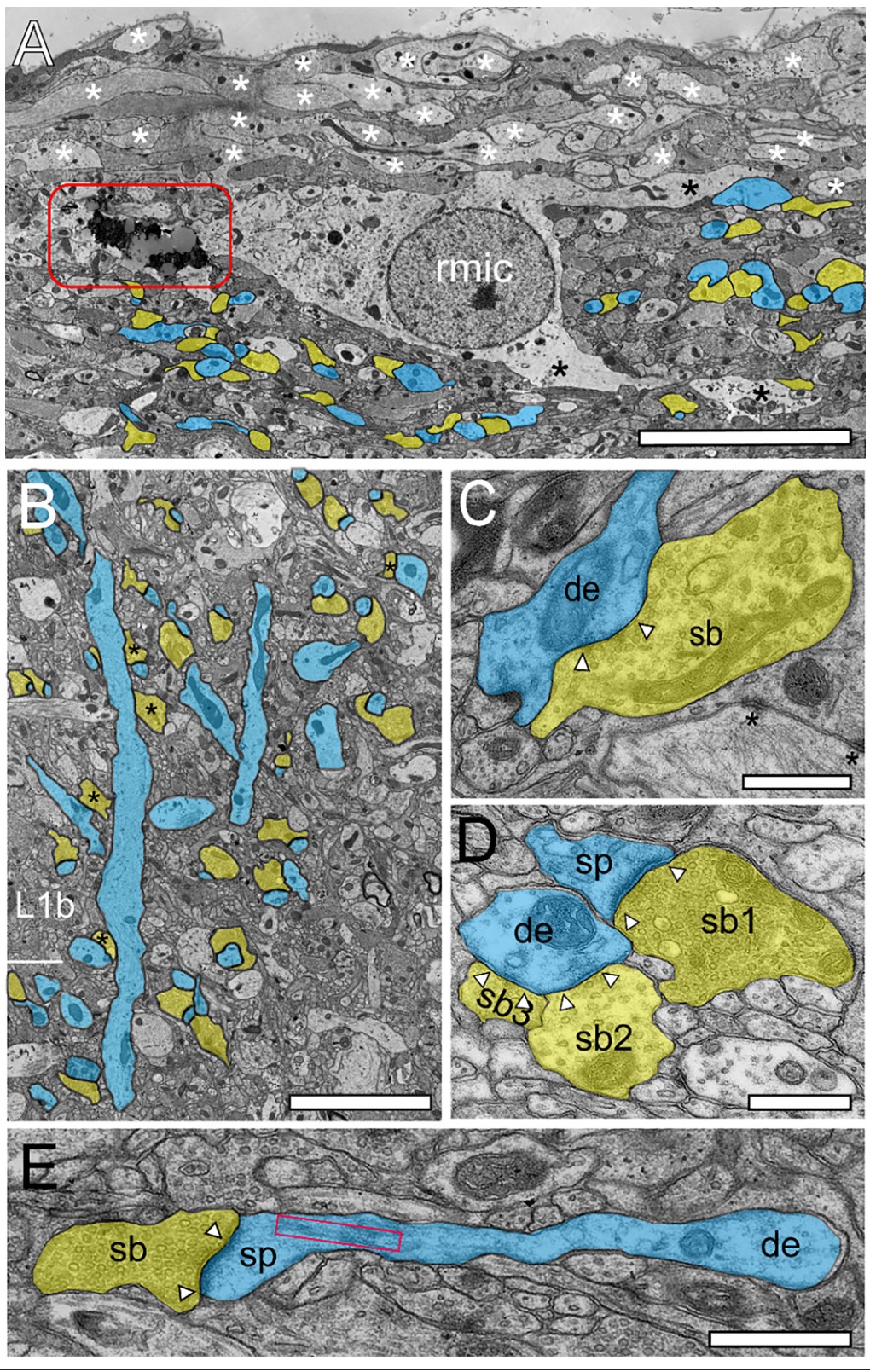

**Figure 2.** Structural and synaptic organization of L1 in the human TLN as revealed by TEM. (**A**) Superficial part of L1a composed of a dense network of fine astrocytic processes (asterisks). In the reactive microglial cell (rmic), so-called lipofuscin granules (framed area) were observed. Below the astrocytic network, synaptic complexes gradually increased (highlighted in blue for target structures, yellow for SBs). Scale bar 10 μm. (**B**) Two apical

*Figure 2 continued on next page*

*Figure 2 continued*

dendrites (blue) traversing L2 with beginning terminal tuft dendrites in L2 and L1 contacted by SBs (transparent yellow); some of which are putatively GABAergic terminating on the dendritic shaft (asterisks). Scale bar 2.5 µm. (**C**) Large GABAergic SB (sb, transparent yellow) identifiable by the small-sized ovoid SVs and the less prominent PSD (arrowheads) synapsing on a dendritic shaft (de, transparent blue) in L1b. (**D**) Three SBs in L1a (sb1-sb3, transparent yellow) located on a dendritic spine (sp, transparent blue) and a neighboring dendritic shaft (sh, transparent blue). Note that sb1 is excitatory as indicated by the larger round SVs with a prominent AZ, whereas sb2 and sb3 are putative inhibitory terminals identified by smaller ovoid SVs and the lack of a prominent PSD. (**E**) Elongated spine (sp) emerging from a small caliber dendrite (de, transparent blue) receiving input from a small-sized SB (sb, transparent yellow). Note the large, elongated spine apparatus (framed area). Scale bar in **C**-**E** 0.5 µm. In graphs **C**-**E**, AZs are marked by arrowheads.

Excitatory and inhibitory synaptic complexes in L1a and L1b were formed by either presynaptic *en passant* or endterminal boutons with their prospective postsynaptic target structures dendritic shafts (*Figures 2B, C, D and 3A–C*) or spines of different caliber and type (*Figures 2B, D, E and 3A–C*). In both sublaminae, SBs were predominantly found on dendritic spines (~81%; L1a) and ~79% (L1b), the remainder were located on dendritic shafts. The majority of SBs on spines were located on mushroom spines (L1a: ~83%, L1b: 73%), a smaller fraction on stubby spines (L1a: ~11%, L1b: ~9%) or on thin elongated spines (L1a: 6%, L1b: ~22%; *Figure 2E*), the remainder were not classifiable.

Since our focus was on excitatory SBs, inhibitory terminals were not further analyzed in detail. Inhibitory SBs were distinguished from excitatory SBs by their smaller SVs, which have a more spherical appearance, a smaller sized PreAZ, and the lack of a prominent PSD (*Figure 2D*). Infrequently, GABAergic and glutamatergic terminals were found on the same dendrite (*Figure 2C and D*) or spine (see for example *Wittner et al., 2001*) but less frequently at ~2–3% (*Kwon et al., 2019*). In turn, not all SBs located on dendritic shafts were GABAergic (see for example *Silver et al., 2003*).

The ratio between excitatory vs. inhibitory SBs was between 10–15% although with layer-specific differences. Our findings are in good agreement with recent publications by *Shapson-Coe et al., 2024* and by *Cano et al., 2023*; *Cano et al., 2024* in which the number of inhibitory SBs in different layers of the temporal lobe neocortex was estimated to be 10–15%. No significant differences were found in the content of inhibitory SBs between the two sublaminae L1a and L1b.

Some SBs in both sublaminae were seen to establish either two or three synapses on the same spine, spines of other origin, or dendritic shafts. Remarkably, ~90% of spines in L1a and L1b contained a spine apparatus (*Figure 2E*), a specialized derivate from the endoplasmic reticulum, structures that increase spine motility and may also stabilize the synaptic complex during signal transduction (*Deller et al., 2003*; reviewed by *Knott and Holtmaat, 2008*).

## Synaptic density in L1a and L1b of the human TLN

Synaptic density measurements serve as a valuable tool for quantitatively characterizing the synaptic organization of a specific cortical layer, assessing the connectivity rate, and identifying potential inter-individual differences among patients within the human TLN.

The average density of synaptic complexes in L1 was relatively high with a value of $5.26*10^8 \pm 8.44*10^7$, $5.52*10^8 \pm 1.25*10^8$ synapses/mm³ in L1a and $5.01*10^8 \pm 1.40*10^7$ synapses/mm³ in L1b, ranging from $4.08*10^8$–$6.31*10^8$ (L1a) and from $4.87*10^8$–$5.15*10^8$ (L1b), respectively (*Supplementary file 1*; *Source data 1*). Strikingly, a huge inter-individual variability was found, although with no significant difference between L1a and L1b.

Therefore, L1 exhibited a relatively high density of synaptic complexes, indicating a relatively robust and thus strong connectivity of neurons within L1 of the human TLN.

## Geometry and size of SBs and mitochondria in the human TLN

Overall, L1 SBs were on average medium-sized, with a mean surface area of 5.48±1.40 µm², and a mean volume of 0.50±0.19 µm³, with a slight difference in size between L1a and L1b. Interestingly, the variability in both surface area and volume of SBs was relatively small in L1a and L1b as indicated by a low CV and variance (*Table 1*; *Source data 2*) regardless of their target structures. SBs in L1 were comparable in size with SBs in L5 and L6, but ~2-fold larger than those in L4 of the human TLN (p≤0.001; see also *Yakoubi et al., 2019a*; *Yakoubi et al., 2019b*; *Schmuhl-Giesen et al., 2022*).

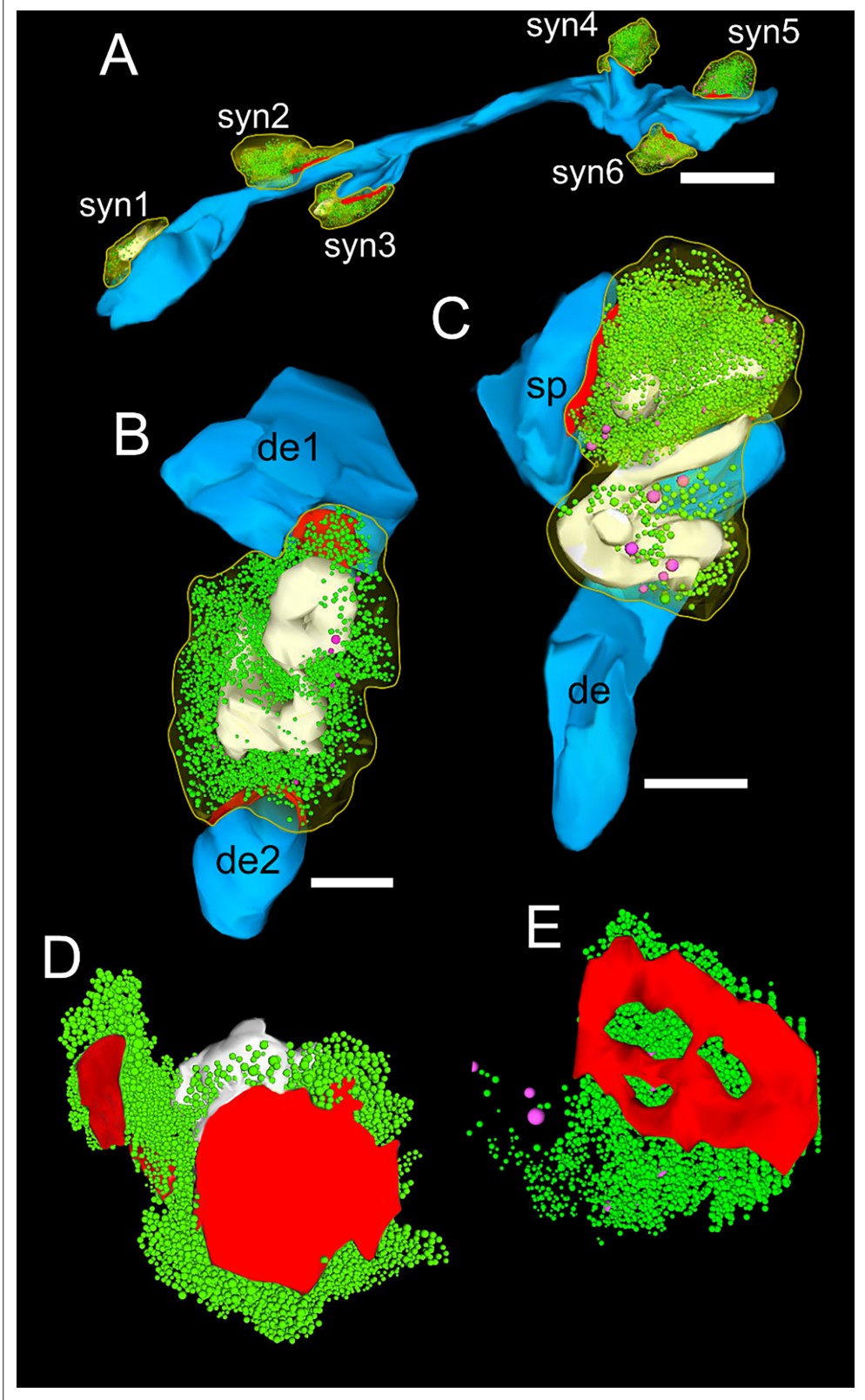

**Figure 3.** 3D-volume reconstructions of SBs and vesicle pools in L1a and L1b of the human TLN. (**A-C**) Here, all SBs are outlined by a yellow contour. Subelements: PreAZs (red), SVs (green), DCVs (magenta), mitochondria (white). The postsynaptic target structures are given in blue. (**A**) Six SBs (sb1-sb6) terminating at different locations on a dendritic segment in L1b, two of which (syn3 and syn4) were located on spines, the remaining on the dendritic

*Figure 3 continued on next page*

*Figure 3 continued*

shaft. Scale bar 1 µm. (**B**) Large SB terminating on two dendritic segments of different shape and size (de1, de2) in L1a. (**C**) SB synapsing on a dendrite (de) and spine (sp) in L1b containing several mitochondria associated with the pool of SVs. (**D, E**) 3D-volume reconstructions of the total pools of SVs (green dots). Note the different shape and size of the PreAZs (red) with either a non-perforated macular (**D**) or perforated (**E**) appearance and the comparably large total pool size. Note the large DCVs intermingled with the pool of SVs.

Mitochondria play a pivotal role in synaptic transmission and plasticity (reviewed by *Dallérac et al., 2018*; see also Discussion). In L1 SBs, either no (*Figures 2B and 3A, E*) or several mitochondria (*Figures 2 and 3C, D*; range 0–8) of different shape and size were observed. When present, mitochondria had a volume of 0.04±0.02 µm$^3$ in L1a and 0.05±0.01 µm$^3$ in L1b, respectively (*Table 1*; *Source data 2*). Mitochondria contributed with ~7% (L1) to the total bouton volume of SBs, similar to values in L6 (~6%; *Schmuhl-Giesen et al., 2022*), but with a ~1.5-fold lower percentage than those in L5 (~12%; *Yakoubi et al., 2019a*) and L4 (~13%; *Yakoubi et al., 2019b*).

A good correlation between the volume of the SBs and that of mitochondria was found for L1a (R$^2$: 0.7528; *Figure 4B*) and for L1b (R$^2$: 0.5992; *Figure 4B*). A weak correlation was also found for L6 (*Schmuhl-Giesen et al., 2022*), but a strong one in both L4 (*Yakoubi et al., 2019a*) and L5 (*Yakoubi et al., 2019b*), suggesting a layer-specific difference in the content of mitochondria in individual SBs.

## Structural composition of AZs in L1a and L1b excitatory SBs in the human TLN

The number, size, and shape of the AZ, composed of the PreAZ and PSD, is one key structural determinant in synaptic transmission and plasticity (*Südhof, 2002*; *Matz et al., 2010*; *Holderith et al., 2012*) as it represents the docking area of SVs (PreAZs) and the receiving area of neurotransmitter quanta at the PSD (*Südhof, 2012*).

The majority (~98%) of SBs in L1a and L1b had only a single (*Figures 2C-E and 3A-C, E*) at most two or three AZs (*Figure 3D*). Besides very large, spanning the entire pre- and postsynaptic apposition zone, also smaller AZs covering only a fraction of the pre- and postsynaptic apposition zone were found.

On average, PreAZs were 0.18±0.08 µm$^2$ in surface area in L1a and 0.22±0.04 µm$^2$ in L1b with only a slight variability between both sublaminae. The surface area of PSDs was quite similar to that of PreAZs, 0.21±0.01 µm$^2$ in L1a and 0.22±0.05 µm$^2$ in L1b, respectively (*Table 1*; *Source data 2*, sheet 01). Thus, the similar size of the PreAZ and PSD showed a nearly perfect overlap at the pre- and postsynaptic apposition zone. L1 AZs did not show a large variability in size as indicated by the low SD, CV, and variance (*Table 1*). Remarkably, AZs exhibited a wide range of shapes in addition to different sizes. Two types of AZs were found: the majority in L1 (~80%) were of the macular and non-perforated type (*Figure 3D*), the remainder being either horseshoe-shaped or ring-like and perforated (*Figure 3E*).

Notably, our analysis revealed only a weak correlation between the surface area of SBs and that of PreAZs for L1a and L1b SBs (*Figure 4A*). These findings imply that the size of SBs and PreAZs is regulated independently.

The size of the synaptic cleft (*Table 1*; *Source data 2*, sheet 03) is an important measure at AZs since its size (diameter) critically determines the temporal and spatial neurotransmitter concentration. Measurements of the size of the synaptic cleft were performed on random TEM images taken from the series using OpenCAR or online directly at the TEM using ImageSP (Fa. Tröndle, Moorenweis, Germany). Only synaptic clefts cut perpendicular to the PreAZ and PSD were included in the sample. The distance between the outer edge of the PreAZ and PSD was measured at the two lateral edges and separately at the central region of the synaptic cleft; the two values of the lateral edges were averaged for each cleft measurement. Finally, a mean ± SD was calculated for both the lateral and central region over all synaptic clefts analyzed per patient.

The size of the synaptic cleft did not differ significantly between the two sublaminae and was similar to the results in L4, L5 (*Yakoubi et al., 2019a*, *Yakoubi et al., 2019b*), and L6 (*Schmuhl-Giesen et al., 2022*).

**Table 1.** Comparative quantitative analysis of various structural and synaptic parameters in L1 of the human TLN.

Summary of different structural parameter measurements (bold) from the detailed 3D-volume reconstructions of SBs in L1, separated for sublaminae. Values are presented as mean, SD, median, IQR, CV, skewness, and variance for each parameter for all patients studied. #: Values with skewness > 3 indicate a non-normal distribution. Abbreviations: IQR: Interquartile Range; CV: coefficient of variation; L: lateral; and C: central (*Source data 2* and *Source data 3*). *Values collected from TEM tomography.

| | Layer | Mean ±SD | Median | IQR | CV | Variance | Skewness |
|---|---|---|---|---|---|---|---|
| | | Synaptic boutons | | | | | |
| **Surface area (µm²)** | L1 | 5.48±1.40 | 5.75 | 2.20 | 0.26 | 1.96 | 0.21 |
| | L1a | 4.58±1.13 | 4.27 | 2.20 | 0.25 | 1.28 | 1.13 |
| | L1b | 6.39±1.08 | 5.87 | 1.97 | 0.17 | 1.17 | 1.66 |
| **Volume (µm³)** | L1 | 0.50±0.19 | 0.38 | 0.33 | 0.39 | 0.04 | 0.13 |
| | L1a | 0.41±0.21 | 0.49 | 0.41 | 0.50 | 0.04 | 0.64 |
| | L1b | 0.58±0.17 | 0.48 | 0.31 | 0.30 | 0.03 | 1.71 |
| | | Active zones | | | | | |
| **PreAZ surface area (µm²)** | L1 | 0.20±0.06 | 0.22 | 0.08 | 0.30 | 0.01 | −1.53 |
| | L1a | 0.18±0.08 | 0.22 | 0.14 | 0.43 | 0.00 | −1.70 |
| | L1b | 0.22±0.04 | 0.22 | 0.08 | 0.18 | 0.00 | 0.00 |
| **PSD surface area (µm²)** | L1 | 0.22±0.07 | 0.23 | 0.12 | 0.32 | 0.00 | −0.91 |
| | L1a | 0.21±0.10 | 0.23 | 0.19 | 0.47 | 0.01 | −1.02 |
| | L1b | 0.22±0.05 | 0.22 | 0.09 | 0.20 | 0.00 | 0.33 |
| **Cleft width (nm)** | L1 | L: 21.75±2.72 C: 29.63±2.36 | L: 22.20 C: 29.73 | L: 2.88 C: 3.19 | L: 0.13 C: 0.08 | L: 7.40 C: 5.56 | L: −1.03 C: −0.96 |
| | L1a | L: 20.17±2.85 C: 29.06±3.48 | L: 21.44 C: 29.17 | L: 5.25 C: 6.95 | L: 0.14 C: 0.12 | L: 8.10 C: 12.08 | L: −1.61 C: −0.14 |
| | L1b | L: 23.34±1.70 C: 30.19±0.92 | L: 22.49 C: 30.19 | L: 3.06 C: 1.84 | L: 0.07 C: 0.03 | L: 2.88 C: 0.85 | L: 1.69 C: 0.00 |
| | | Mitochondria | | | | | |
| **Volume (µm³)** | L1 | 0.04±0.02 | 0.04 | 0.03 | 0.35 | 0.00 | −0.31 |
| | L1a | 0.04±0.02 | 0.04 | 0.04 | 0.50 | 0.00 | 0.00 |
| | L1b | 0.05±0.01 | 0.04 | 0.02 | 0.25 | 0.00 | 1.73 |
| **% to the total volume** | L1 | 7.21±1.10 | 7.56 | 1.81 | 0.15 | 1.21 | −1.13 |
| | L1a | 7.18±1.60 | 8.06 | 2.81 | 0.22 | 2.56 | −1.73 |
| | L1b | 7.23±0.68 | 7.17 | 1.35 | 0.09 | 0.46 | 0.42 |
| | | Synaptic vesicles | | | | | |
| **Total number of SVs** | L1 | 3430.97±1773.77 | 3675.96 | 3386.86 | 0.52 | 3146276.00 | −0.31 |
| | L1a | 2958.62±1940.51 | 2980.76 | 3880.83 | 0.66 | 3765578.00 | −0.05 |
| | L1b | 3903.32±1852.23 | 4371.16 | 3614.75 | 0.47 | 3430758.00 | −1.06 |
| **Volume (µm³)** | L1 | 0.03±0.02 | 0.03 | 0.03 | 0.60 | 0.00 | 0.94 |
| | L1a | 0.02±0.01 | 0.02 | 0.02 | 0.50 | 0.00 | 0.00 |
| | L1b | 0.04±0.02 | 0.04 | 0.04 | 0.50 | 0.00 | 0.00 |

*Table 1 continued on next page*

*Table 1 continued*

|  | Layer | Mean ±SD | Median | IQR | CV | Variance | Skewness |
|---|---|---|---|---|---|---|---|
| Vesicle diameter (nm) | L1 | 25.03±5.67 | 26.36 | 10.38 | 0.23 | 32.13 | –0.31 |
|  | L1a | 23.33±5.27 | 25.88 | 9.56 | 0.23 | 27.74 | –1.67 |
|  | L1b | 26.73±6.63 | 28.25 | 12.99 | 0.25 | 43.93 | –0.98 |
|  | L1 * | 30.29±4.67 | 31.10 | 3.21 | 0.15 | 21.77 | –2.60 |
|  | L1a* | 28.11±6.10 | 30.15 | 3.91 | 0.22 | 37.17 | –2.04 |
|  | L1b* | 31.88±2.36 | 31.62 | 3.27 | 0.07 | 5.56 | 0.04 |
|  | Pool sizes of SVs |  |  |  |  |  |  |
| Putative RRP p10 nm from the PreAZ | L1 | 3.60±4.24 | 2.10 | 6.98 | 1.18 | 17.99 | 1.17 |
|  | L1a | 5.90±5.05 | 6.11 | 10.10 | 0.86 | 25.53 | –0.18 |
|  | L1b | 1.29±1.87 | 0.41 | 3.41 | 1.44 | 3.49 | 1.65 |
| Putative RRP p20 nm from the PreAZ | L1 | 19.05±17.23 | 17.17 | 29.37 | 0.90 | 296.86 | 0.67 |
|  | L1a | 25.04±21.09 | 24.82 | 42.18 | 0.84 | 444.82 | 0.05 |
|  | L1b | 13.07±13.78 | 9.52 | 26.86 | 1.05 | 189.80 | 1.08 |
| Putative RP 60–200 nm from the PreAZ | L1 | 463.00±283.82 | 512.65 | 554.39 | 0.61 | 80553.57 | –0.19 |
|  | L1a | 390.12±286.89 | 335.05 | 565.79 | 0.74 | 82304.11 | 0.83 |
|  | L1b | 535.88±321.16 | 690.25 | 584.03 | 0.60 | 103145.30 | –1.66 |
| Putative resting pool >200 nm from the PreAZ | L1 | 2896.50±1435.93 | 3120.39 | 2698.41 | 0.50 | 2061880.00 | –0.38 |
|  | L1a | 2515.02±1588.99 | 2662.80 | 3167.65 | 0.63 | 2524880.00 | –0.41 |
|  | L1b | 3277.97±1480.96 | 3577.98 | 2915.99 | 0.45 | 2193254.00 | –0.87 |

## Organization of the pools of SVs in L1a and L1b excitatory SBs of the human TLN

SVs are the other key structure in neurotransmitter storage and release, hence they play a fundamental role in synaptic transmission and in the modulation of short- and long-term synaptic plasticity (*Südhof, 2002*; *Südhof, 2012*). Three different pools of SVs are functionally defined: the RRP, the RP, and the resting pool. Synaptic efficacy, strength, mode, and probability of release ($P_r$) are regulated by these pools (*Schikorski and Stevens, 2001*; *Silver et al., 2003*; *Rizzoli and Betz, 2004*; *Saviane and Silver, 2006*; *Schikorski, 2014*; *Watanabe et al., 2014*; *Neher, 2015*; *Vaden et al., 2019*; reviewed by *Rizzoli and Betz, 2005*; *Denker and Rizzoli, 2010*; *Chamberland and Tóth, 2016*).

In general, SVs were distributed throughout the entire terminal in ~95% of the population of SBs investigated (*Figures 2 and 3*), while the remainder showed a more cluster-like distribution. Two different types of vesicles were found: (1) Small clear SVs with an average diameter of 25.03±5.27 nm ranging up to ~40 nm (*Table 1*; *Source data 2*, sheet 01). Using TEM tomography, the SV diameter was on average 30.29±4.67 nm ranging up to ~47 nm (n=38 SBs with a total of 5.834 SVs; *Table 1*; *Source data 3*, sheet 05). (2) DCVs (Dense Core Vesicles; *Figure 3B, C, E*; *Source data 2*, sheet 01) with an average diameter of 57.97±6.45 nm with no significant difference between both sublaminae. DCVs were intermingled with the population of SVs throughout the entire SB (*Figure 3B, C, E*). DCVs play an important role in endo- and exocytosis by releasing various co-transmitters, such as neuropeptides, ATP, noradrenalin, and dynorphin (*Ghijsen and Leenders, 2005*) or in synaptogenesis (*Sorra et al., 2006*) and the build-up of PreAZs by releasing Piccolo and Bassoon (*Zhai et al., 2001*; *Shapira et al., 2003*). In addition, they are involved in clustering SVs at the PreAZs (*Mukherjee et al., 2010*; *Watanabe et al., 2014*).

The distribution pattern of SVs made it impossible to morphologically distinguish between the three functionally defined pools of SVs, with the exception of the so-called 'docked' ones, that are fused to the PreAZ, and the omega-shaped bodies, that have already opened and released quanta

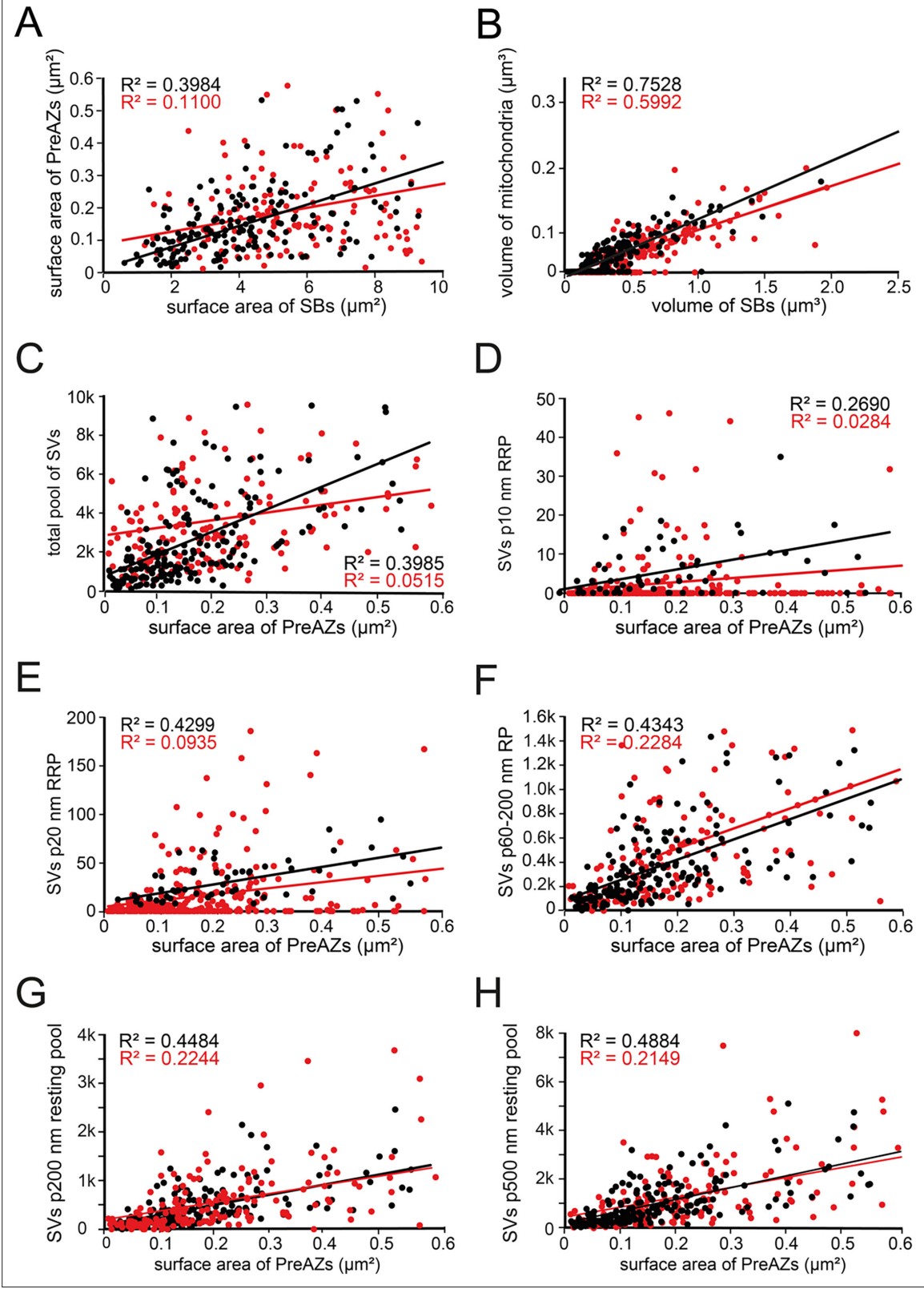

**Figure 4.** Correlation plots for structural and synaptic parameters characterizing L1 SBs. Correlation plots showing the strength of correlations between structural and synaptic parameters (*Source data 2*). Correlations dots and regression lines for L1a are given in black and that for L1b in red. (**A**) Surface area of SBs vs. surface area of PreAZs; (**B**) Volume of SBs vs. volume of mitochondria; (**C**) Surface area of PreAZs vs. total pool of SVs; (**D**) Surface area of PreAZs vs. p10 nm RRP; (**E**) Surface area of PreAZs vs. p20 nm RRP; (**F**) Surface area of PreAZs vs. p60-p200 nm RP; (**G**) Surface area of PreAZs vs. p200 nm resting pool; (**H**) Surface area of PreAZs vs. p500 nm resting pool.

of neurotransmitter (*Rizzoli and Betz, 2004*; *Denker and Rizzoli, 2010*; *Neher, 2015*; reviewed by *Rizzoli and Betz, 2005*; *Chamberland and Tóth, 2016*). However, an attempt was made to define a morphological correlate for the three functionally defined pools.

For this purpose, a perimeter analysis (a modified Sholl analysis) of each SV in individual SBs was carried out. The perimeter was defined as the minimum distance between each SV membrane and the PreAZ membrane (*Sätzler et al., 2002*; for criteria to define SV pools see also *Rizzoli and Betz, 2005*; *Denker and Rizzoli, 2010*). The putative RRP was defined as a perimeter (p) $P \leq 10$ nm and $P \leq 20$ nm to the PreAZ. These criteria for the RRP from which SVs could be easily and rapidly recruited were chosen because both values are less than a SV diameter apart from the PreAZ. The putative RP maintains release during moderate stimulation and was classified at a distance from the PreAZ between p60-p200 nm. All SVs that were further than 200 nm from the PreAZ were regarded as constituting the putative resting pool, which acts as a deposit of SVs only used by intense and/or repetitive stimulation.

On average, the total pool of SVs was 2958.62±1940.51 in L1a vs. 3903.32±1852.23 in L1b with an average of 3430.97±1773.77 for L1 (*Table 1*; *Source data 2*). Remarkably, the total pool in L1b was significantly larger (p≤0.01) by ~1.3-fold when compared to L1a. In both sublaminae (*Figure 3D, E*), there is huge variability in total pool size, ranging from 758 to 6542 in L1a and 1507–15,060 in L1b, as indicated by the SD, IQR, and variance (*Table 1*). SVs contributed with ~5% (0.03 µm$^3$) in L1a and ~4% (0.02 µm$^3$) in L1b to the total volume of SBs. However, no significant difference between both sublaminae was observed (*Table 1*; *Source data 2*).

Interestingly, no correlation between the volume of SBs vs. the total pool size was found for L1a ($R^2$: 0.2490), but a good correlation for L1b ($R^2$: 0.5700). Furthermore, no correlation was found between the PreAZ vs. the total pool size (*Figure 4C*; *Source data 2*).

The average number of SVs in the putative RRP/PreAZ for L1 was 3.60±4.24 SVs at p10 nm (L1a: 5.90±5.05 SVs and L1b: 1.29±1.87 SVs) with a significant difference (p≤0.01) by ~4.5-fold between the two sublaminae (*Table 1*; *Source data 2*). The putative RRP/PreAZ at the p20 nm criterion was on average 19.05±17.23 SVs (L1a: 25.04±21.09 SVs and L1b: 13.07±13.87 SVs) and thus nearly 2-fold larger for L1a, although with a huge variability, but even though it may point to sublaminae-specific differences in $P_r$, synaptic efficacy, strength, and paired-pulse behavior at individual SBs.

No correlation between the putative RRP at both the p10 nm and p20 nm criterion and the surface area of PreAZs was found for both sublaminae (*Figure 4D and E*).

The putative RP/PreAZ at the 60–200 nm perimeter criterion was 390.12±286.89 SVs in L1a and 535.88±321.16 in L1b with no significant difference between both sublaminae (*Table 1*; *Source data 2*, sheet 01). It should be noted, however, that the variability of the putative RP is relatively large in both sublaminae as indicated by the SD, median, IQR, and variance (*Table 1*).

Again, no correlation was observed between the SVs in the putative p60-200 nm RP and the surface area of the PreAZs in both L1a (*Figure 4F*) and L1b (*Figure 4F*; *Source data 2*, sheet 01).

The putative resting pool of SVs contained on average 2515.02±1588.99 SVs in L1a and 3277.97±1480.96 SVs in L1b with a significant difference between the two sublaminae (p≤0.01, *Table 1*; *Source data 2*, sheet 01). The number of SVs in the putative resting pool in L1b was about 1.3-fold larger when compared to L1a. Again, also the putative resting pool showed a large variability for both sublaminae as indicated by the SD, CV, IQR, and variance (*Table 1*).

No correlation was observed between the SVs in the putative p200 nm and p500 nm resting pool and the surface area of the PreAZs in L1a and L1b (*Figure 4G and H*; *Source data 2*, sheet 01).

Neither for the total nor for the three different putative pools of SVs were significant differences between SBs terminating on dendritic spines or shafts found (*Source data 2*, sheet 04).

## TEM tomography of L1 excitatory SBs in the human TLN

It is still controversially discussed whether so-called 'docked' SVs or omega-shaped bodies represent the RRP. To compare our results from the perimeter analysis for the p10 nm RRP with that of 'docked' SVs, high-resolution TEM tomography was carried out. Only SBs where the AZ could be followed from its beginning to its end in individual tilt-series and where the AZ was cut perpendicular through the PreAZ, PSD and the synaptic cleft were analyzed. In L1a (50 SBs, on dendritic shafts = 25 SBs; on spines = 25 SBs) and L1b (50 SBs, on dendritic shafts = 25 SBs; on spines = 25 SBs) the number of 'docked' SVs was analyzed at the PreAZ (*Figure 5*; *Table 2*; *Source data 3*).

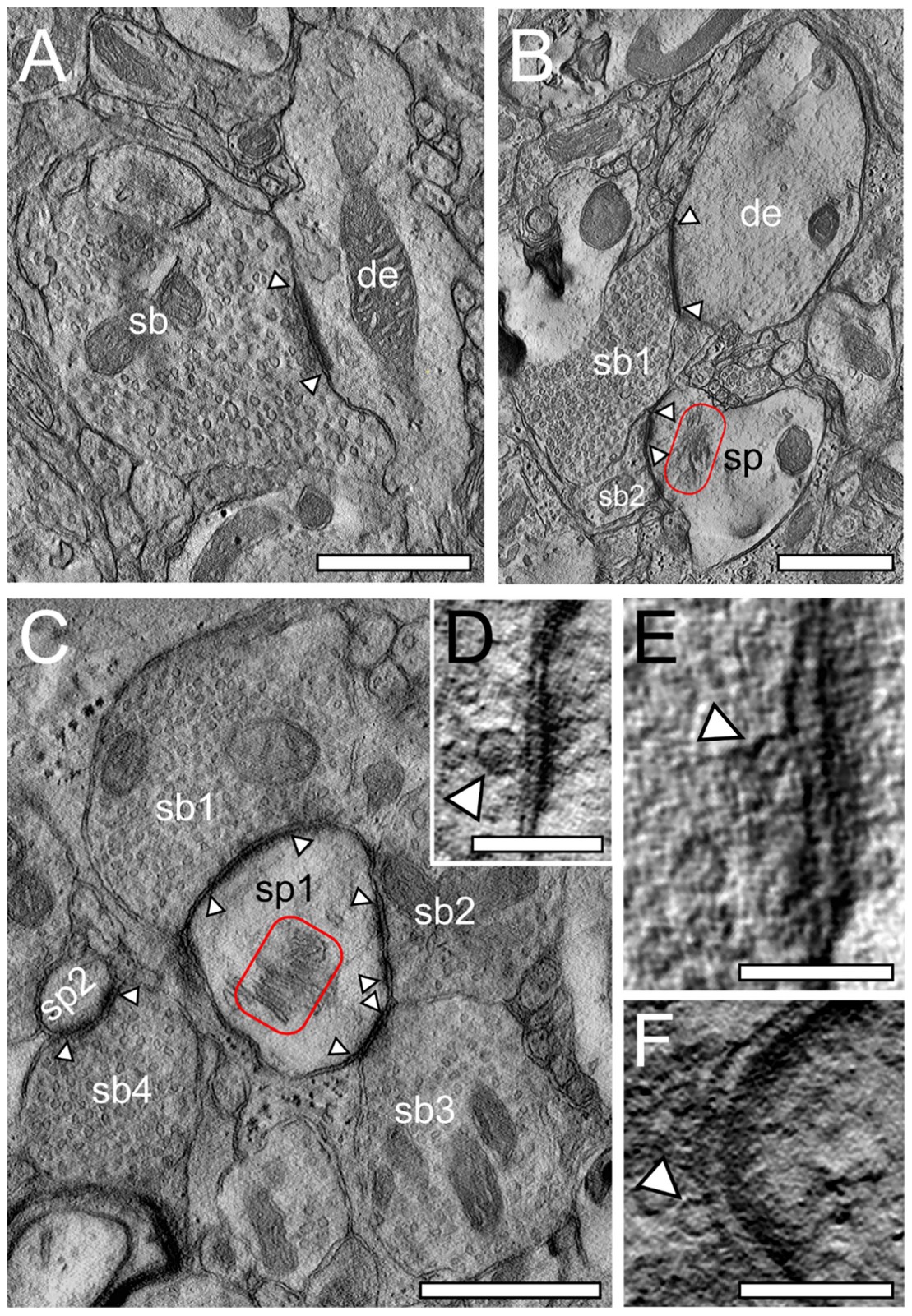

**Figure 5.** TEM tomography of SBs in L1a and L1b in the human TLN. (**A**) Large SB (sb) terminating with a single AZ (arrowheads) on a terminal tuft dendritic shaft (de) in L1a. (**B**) Two adjacent SBs synapsing either (sb1) on a thick dendritic segment (de) or (sb2) on a stubby spine (sp) with a prominent spine apparatus (framed area) in L1b. Scale bars 1 μm. (**C**) Large mushroom (sp1) and a small spine head (sp2) receiving input from three SBs (sb1, sb2, sb3) and a single SB (sb4) in L1b. Note the large AZs (arrowheads) and the prominent spine apparatus (red framed area) in the mushroom spine. Scale

*Figure 5 continued on next page*

Figure 5 continued

bar 0.5 µm. (**D**, **E**, **F**) High-power images of 'docked' SVs (arrowheads) taken from a tilt-series through an individual PreAZ at a L1a spine SB. Note the so-called omega-shaped bodies in (**E**, **F**) pointing to the already opening and release of glutamate quanta. Scale bars 0.1 µm.

The results for L1a and L1b were two-fold: First, in agreement with already published results (*Yakoubi et al., 2019a*; *Yakoubi et al., 2019b*; *Schmuhl-Giesen et al., 2022*) only in a minority (<1%) of all PreAZs analyzed, regardless of their target structures, a dendritic shaft (*Figure 5A*) or spine (*Figure 5B and C*), no 'docked' SVs were observed, which is in contrast to our perimeter analyses showing several boutons without vesicles within the p10 nm criterion. Second, the majority of PreAZs (~98%) contained more than 2, the most 8 (L1a) and 6 (L1b) 'docked' SVs. This finding strongly supports multivesicular release of 'docked' SVs in L1a and L1b SBs in line with findings in L4, L5, and L6 SBs (*Yakoubi et al., 2019a*; *Yakoubi et al., 2019b*; *Schmuhl-Giesen et al., 2022*; see also *Figure 5D–F*).

On average, 3.71±1.38 'docked' SVs (L1a) and 3.42±1.34 'docked SVs' (L1b) were found at individual PreAZs with similar values for both sublaminae. Furthermore, no significant difference was found for 'docked' SVs between shaft vs. spine synapses in L1. The number of 'docked' SVs at L1a PreAZs was ~2-fold smaller but nearly 3-fold larger in L1b when compared to the results of our quantitative perimeter analysis for the putative RRP using the p10 nm criterion. In contrast to the p10 nm criterion with high inter- and intra-individual variabilities (Appendix 2), the number of 'docked' SVs showed relatively low variability as indicated by the SD, CV, variance, and skewness. They were different (p≤0.001) from the values estimated with the p10 nm perimeter analysis.

There seems to be a tendency that larger PreAZs contained more 'docked' SVs, providing a larger 'docking' area allowing the recruitment of more SVs.

In summary, a notable disparity difference was observed between values obtained for L1 using the p10 nm criterion of the perimeter and the TEM tomography analysis. Additionally, the putative RRP in L1 (this study) and L6 (*Schmuhl-Giesen et al., 2022*) was ~2–4 times smaller when compared to values in L4 (*Yakoubi et al., 2019b*) and L5 (*Yakoubi et al., 2019a*), pointing toward a layer-specific regulation of the putative RRP. Moreover, the results suggest that not only 'docked' SVs but also those very close to the PreAZ should be considered to belong to the putative RRP.

## Astrocytic coverage of L1 SBs in the human TLN

Astrocytes, by directly interacting with synaptic complexes thus forming the 'tripartite' synapse, play a pivotal role in the induction, maintenance, and termination of synaptic transmission by controlling the spatial and temporal concentration of neurotransmitter quanta in the synaptic cleft (reviewed by *Dallérac et al., 2018*). Astrocytic profiles were identified by their irregular stellate shape, relatively clear cytoplasm, numerous glycogen granules, and bundles of intermediate filaments (for criteria see for example *Peters et al., 1991*; *Ventura and Harris, 1999*; see also *Figure 6A and B*). Astrocytes and their fine processes formed a relatively dense network in L4 and L5, but a relatively loose one within the neuropil in L1, as determined by measuring the volume content of astrocytic processes. Astrocytic processes in L1 contributed by ~20% to the total volume of the human TLN, which is similar

**Table 2.** 'Docked SVs' in L1a and L1b of the human TLN.
Summary of the number of 'docked' SVs in L1, separately for both sublaminae as well as for the different target structures dendritic shafts vs. dendritic spines. Mean ± SD, Median, Interquartile Range (IQR), coefficient of variation (CV), Skewness, and Variance are given for each parameter in all patients studied (*Source data 3*).

| | Layer | Mean ±SD | Median | IQR | CV | Skewness | Variance |
|---|---|---|---|---|---|---|---|
| | L1 (n=360 SVs) | 3.56±1.36 | 4.00 | 3.00 | 0.38 | –0.07 | 1.85 |
| | L1a | 3.71±1.38 | 4.00 | 3.00 | 0.37 | –0.16 | 1.89 |
| | SBs on dendritic shafts (n=91 SVs) | 3.64±1.47 | 4.00 | 3.00 | 0.40 | –0.08 | 2.16 |
| | SBs on dendritic spines (n=98 SVs) | 3.77±1.31 | 4.00 | 2.00 | 0.35 | –0.24 | 1.70 |
| | L1b | 3.42±1.34 | 3.00 | 3.00 | 0.39 | 0.02 | 1.80 |
| | SBs on dendritic shafts (n=87 SVs) | 3.48±1.45 | 3.00 | 3.00 | 0.42 | 0.04 | 2.09 |
| Number of docked vesicles | SBs on dendritic spines (n=84 SVs) | 3.36±1.25 | 3.00 | 2.50 | 0.37 | –0.07 | 1.57 |

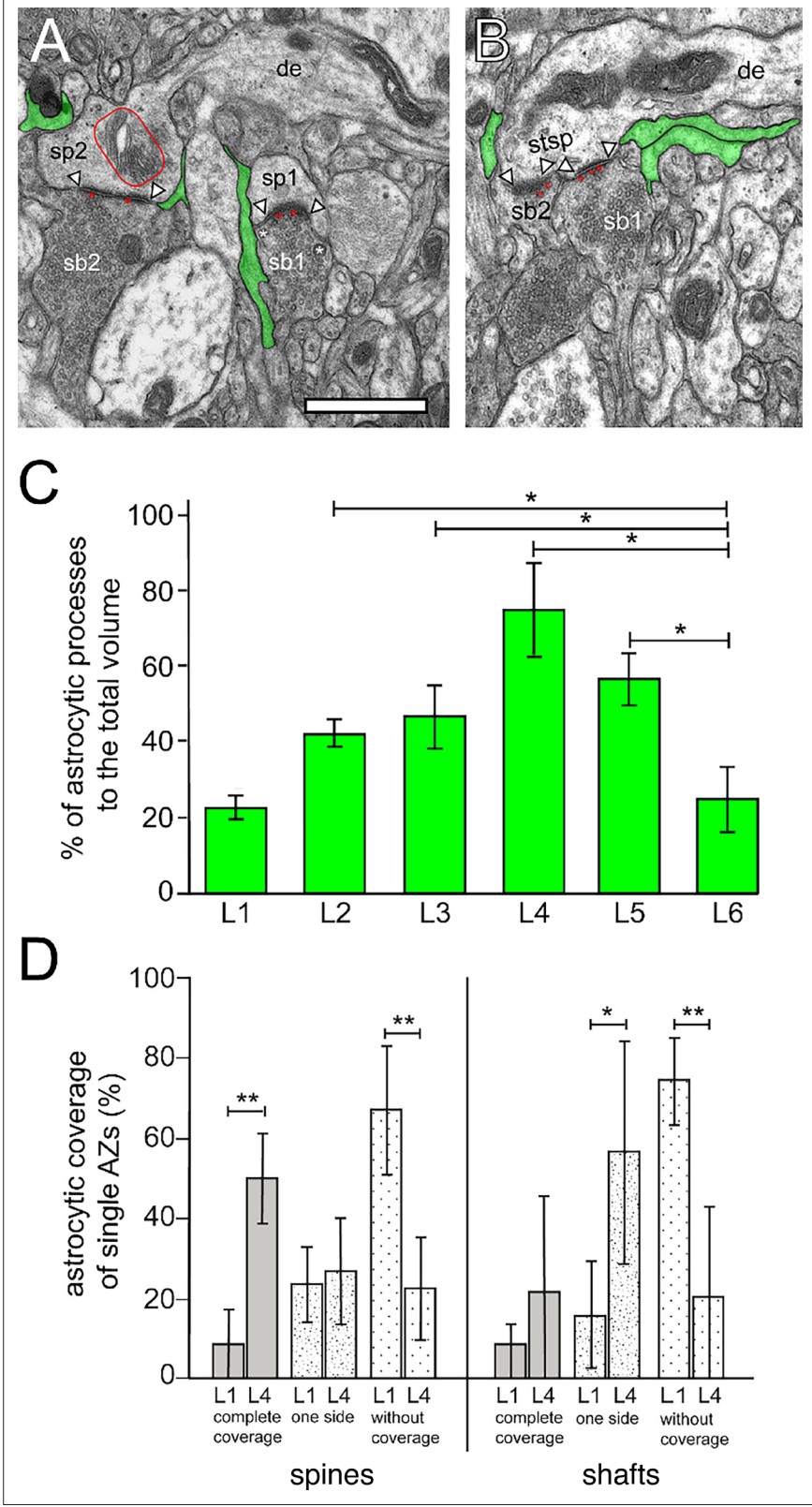

**Figure 6.** Astrocytic coverage of synaptic complexes in the human TLN. (**A**) TEM micrograph of two adjacent SBs (sb1, sb2) terminating on two spines (sp1, sp2) in L1a. Sp2 contained a prominent spine apparatus (framed area), sb1 two DCVs (asterisks). Note that both synaptic complexes were only partially ensheathed by fine astrocytic processes (transparent green) reaching the AZs only on one side. In both synaptic complexes, two 'docked' SVs

*Figure 6 continued on next page*

*Figure 6 continued*

(transparent red) at the PreAZs were found. Scale 0.25 µm. (**B**) Stubby spine (stsp) receiving two SBs (sb1, sb2) in L1b. Here, fine astrocytic processes (transparent green) were found close to the two synaptic complexes but never reached the synaptic cleft. Note also the 'docked' SVs (transparent red) pointing to multivesicular release. In both images, the AZs are marked by arrowheads. Scale bar 0.25 µm. (**C**) Bar histogram showing the percentage (mean ± SD) of the volumetric fraction of astrocytic processes to the total volume in L1 to L6 (*Source data 4*, sheet 01). Values for L4-L6 are taken from *Yakoubi et al., 2019a*; *Yakoubi et al., 2019b*; *Schmuhl-Giesen et al., 2022*. (**D**) Bar histogram showing the percentage (mean ± SD) of AZs covered by fine astrocytic processes, either on both sides, on one side, or without any coverage, subdivided by their location on dendritic spines or shafts in L1 and L4 (*Source data 4*, sheet 02). The horizontal bars in C and D indicate significant differences *p<0.05; **p<0.01.

to findings in L6 of the human TLN. The highest degree (~70%) of astrocytic processes was measured in L4 (*Figure 6C*; *Source data 4*, sheet 01, *Table 3*).

A second, more detailed measurement (*Figure 6D*, *Table 3*; *Source data 4*, sheet 02), in which the percentage of AZs covered by astrocytes was analyzed, demonstrated that (1) in L1, the majority of spine and shaft synapses were not (~70%) or only partially (~20%) covered. by fine astrocytic processes; (2) only in a small proportion (~10%) at both spine and shaft synapses fine astrocytic processes forming a 'tripartite' synapse physically isolating the synaptic complexes from the surrounding neuropil were observed; (3) the results in L1 (L1a: n=3 patients, n=128 spine synapses, n=48 shaft synapses; L1b: n=3 patients, n=126 spine synapses, n=44 shaft synapses) are in contrast to those in L4 (n=5 patients; n=93 spine synapses, n=41 shaft synapses). In L4, a much higher number of 'tripartite' synaptic complexes (spines: ~50%; shafts: ~20%) was found. In this layer, ~25% of the AZs on dendritic spines and ~60% on dendritic shafts were partially covered, however, with large inter-individual differences. Approximately 20% of both spine and shaft synapses were not covered; (4) no significant differences between spine and shaft synapses were found in either L1 or L4.

In summary, it is most likely that the partial coverage or complete absence of fine astrocytic processes at the majority of synaptic complexes in L1 of the human TLN may contribute to different 'behaviors' of synaptic complexes in synaptic transmission and in the modulation of short-term plasticity, for

**Table 3.** Astrocytic coverage of synaptic complexes in the human TLN.

Summary of astrocytic coverage of synaptic complexes in the human TLN in L1 and L4 measured with two independent experimental approaches: (1) percentage of astrocytic processes contributing to the total volume and (2) percentage of astrocytic coverage around individual AZs.

| Layer / Measurement | L1 (Mean ±SD) | L2 (Mean ±SD) | L3 (Mean ±SD) | L4 (Mean ±SD) | L5 (Mean ±SD) | L6 (Mean ±SD) |
|---|---|---|---|---|---|---|
| Percentage of astrocytic processes to the total volume | | | | | | |
| | 23.23±6.90 | 43.10±3.38 | 48.42±10.06 | 76.71±12.94 | 58.13±9.15 | 25.16±9.56 |
| Astrocytic coverage of single AZs (%) | | | | | | |
| Spine synapses | n=254 (L1a: 128; L1b: 126) | — | — | n=93 | — | — |
| Complete coverage | 8.69±9.10 | | | 50.21±11.21 | | |
| One side | 23.73±9.29 | | | 27.26±13.04 | | |
| Without coverage | 67.57±15.78 | | | 22.52±12.77 | | |
| Shaft synapses | n=92 (L1a: 48; L1b: 44) | — | — | n=41 | — | — |
| Complete coverage | 8.54±5.39 | | | 22.09±24.16 | | |
| One side | 16.44±13.16 | | | 57.30±27.80 | | |
| Without coverage | 75.02±10.89 | | | 20.61±22.23 | | |

example in the removal of 'spilled' horizontally diffusing neurotransmitter quanta favoring synaptic crosstalk (for more detail see Discussion).

## Discussion

This study investigated the synaptic organization of L1 in the human TLN. In addition to similarities, the quantitative 3D models of the analyzed excitatory SBs showed significant layer-specific differences in structural and synaptic parameters, in particular in the size of AZs and the total pool of SVs, the putative RRP, RP, and resting pool compared to other previously studied layers (*Yakoubi et al., 2019a*; *Yakoubi et al., 2019b*; *Schmuhl-Giesen et al., 2022*). The low degree of astrocytic coverage of L1 SBs suggests that glutamate spillover, and as a consequence, synaptic cross talk may occur at most synaptic complexes in L1.

### Synaptic density measurements

Synaptic density measurements are a useful tool to describe the synaptic organization of a particular area, nuclei, and even layers in different brain regions, but also the degree of connectivity underlying the computational properties of a given brain area or in a given brain network. Meanwhile, numerous studies in various animal species and brain regions have performed such an analysis, but data for a density of synaptic complexes in humans are still rare (but see *Marco and DeFelipe, 1997*; *Tang et al., 2001*; *DeFelipe et al., 1999*; *DeFelipe et al., 2002*; *Alonso-Nanclares et al., 2008*; *Blazquez-Llorca et al., 2013*; *Finnema et al., 2016*; *Cano et al., 2021*; *Cano et al., 2023*).

Strikingly, a huge layer-specific difference in the mean density of synaptic contacts was found in the human TLN (see also *Supplementary file 1*). In L1, the overall density was $5.26*10^8 \pm 8.44*10^7$ synaptic complexes/mm$^3$, although a great variability was observed as indicated by the SD, consistent with findings in the other layers of the human TLN L4 ($2.37*10^6 \pm 2.19*10^6$; *Yakoubi et al., 2019b*); L5 ($3.89*10^8 \pm 9.12*10^8$; *Yakoubi, 2023*); and L6 ($4.98*10^7 \pm 1.85*10^7$; *Schmuhl-Giesen et al., 2022*) and existing data published by *Alonso-Nanclares et al., 2008*: $9.13 \pm 0.63*10^8$ and *Tang et al., 2001*: $164*10^{12}$. The density values for L1 and L5 are in the same order of magnitude but differ substantially from L4 and L6 by one or two orders of magnitude. The huge differences in synaptic density may be explained by gender and age-specific differences and the use of different methods in estimating synaptic densities (*DeFelipe et al., 1999*).

In summary, the highest synaptic density was observed in L1, indicating a relatively high level of connectivity and synaptic interaction. This may suggest that L1 facilitates rapid information processing due to its robust synaptic activity.

### Important structural subelements of SBs in the human TLN

#### Shape and size of PreAZs and PSDs

Synaptic efficacy, strength, modes of release, and $P_r$ are beside the pool of SVs, determined by the shape and size of AZs (*Matz et al., 2010*; *Holderith et al., 2012*; *Südhof, 2012*).

The majority of SBs in L1 of the human TLN had a single, at most three AZs, that could be of the non-perforated macular or perforated type. These findings are comparable with results for other layers in the human TLN (L4: *Yakoubi et al., 2019b*, L5: *Yakoubi et al., 2019a*, L6: *Schmuhl-Giesen et al., 2022*-) but by ~1.5-fold larger than in rodent and non-human primate neocortex (*Marrone et al., 2005*; *Rollenhagen et al., 2015*; *Rollenhagen et al., 2018*; *Bopp et al., 2017*; *Hsu et al., 2017*).

The surface area of AZs in L1 SBs of the human TLN was on average ~0.20 μm$^2$. This is in good agreement with data obtained for AZs in L3 (*Cano et al., 2021*), L4 (*Yakoubi et al., 2019b*), L5 (*Yakoubi et al., 2019a*), L6 (*Schmuhl-Giesen et al., 2022*), and in L3 (*Cano et al., 2023*) of the human temporal and cingulate neocortex. However, they were ~2- to 3-fold larger than those in mouse and non-human primates visual, motor, and somatosensory neocortex (*Bopp et al., 2017*; *Hsu et al., 2017*) but remarkably even larger than AZs in comparably large CNS terminals like the hippocampal mossy fiber bouton (*Rollenhagen et al., 2007*), the cerebellar mossy fiber bouton (*Xu-Friedman and Regehr, 2003*), and the Calyx of Held in the medial nucleus of the trapezoid body (*Sätzler et al., 2002*).

The substantial variability in the size of AZs at individual SBs, observed both in the human TLN and in experimental animals, likely contributes to differences in synaptic efficacy, strength, $P_r$, quantal size, as well as the size of the RRP and RP (*Südhof, 2002*; *Matz et al., 2010*; *Freche et al., 2011*; *Holderith et al., 2012*; *Neher, 2015*; *Chamberland and Tóth, 2016*; *Rollenhagen et al., 2018*; *Vaden et al., 2019*; reviewed by *Rizzoli and Betz, 2005*; *Denker and Rizzoli, 2010*). However, it has to be noted that the size of the AZ is regulated as a function of activity, as shown for hippocampal SBs in the CA1 subregion (*Matz et al., 2010*; *Holderith et al., 2012*). However, the comparably large size of the AZs in the human TLN also suggests a larger 'docking' area allowing the fusion of more SVs and thus a larger $P_r$.

It has been postulated that perforated AZs contribute to synaptic efficacy and plasticity (*Geinisman et al., 1992*; *Nava et al., 2014*) by an increased number of SVs at perforated synapses (*Buchs and Muller, 1996*; *Desmond and Weinberg, 1998*) and by increasing the number of AMPA- and NMDA-type glutamate receptors at perforated synapses (*Ganeshina et al., 2004*). However, since the number of perforated AZs in all layers of the human TLN is relatively low compared to those of the macular-non-perforated type, it is highly speculative which impact they might have here on synaptic efficacy and the modulation of synaptic plasticity.

## Size of SVs obtained with TEM vs. TEM tomography

The values obtained with both TEM (25.03±5.27 nm) and TEM tomography (30.29±4.67 nm) in L1 are within a range published in previous studies (e.g. human TLN: *Yakoubi et al., 2019a*, *Yakoubi et al., 2019b*; *Schmuhl-Giesen et al., 2022*; rat hippocampal CA3 region: *Rollenhagen et al., 2007*; *Zhao et al., 2012a*; rodent hippocampal CA1 region: *Harris and Sultan, 1995* [lower range]; rodent somatosensory cortex: *Rollenhagen et al., 2015*; *Rollenhagen et al., 2018*; *Prume et al., 2020*). Based on TEM studies, also higher mean SV diameters of ~40 nm and beyond have been published for different species and brain regions (e.g. *Schikorski and Stevens, 1997*; *Sätzler et al., 2002*; *Deák et al., 2004*; *Hu et al., 2008*). TEM tomographic analyses also indicate variability in SV diameter from ~30 to~52 nm in different brain regions and mouse knock-out model systems (*Imig et al., 2014*; *Muth et al., 2024*). High variability of SV diameter has been reported to be a general phenomenon in several species and brain regions. Even a synapse-to-synapse variability in mean SV size is described in neighboring synapses in the hippocampus, neocortex, and cerebellum of rats, mice, and human cell culture systems (*Hu et al., 2008*; *Qu et al., 2009*). Additionally, within a single SB, the size of individual SVs can vary, for example, depending on their transmitter filling state and content (e.g. *Colliver et al., 2000*; *Pothos et al., 2002*).

## Size of the three pools of SVs

Besides the size of the PreAZ, the pool of rapidly releasable SVs also critically determines $P_r$ and thus synaptic efficacy, strength, and plasticity (*Rosenmund and Stevens, 1996*; *Schikorski and Stevens, 2001*; *Rizzoli and Betz, 2004*; *Schikorski, 2014*; *Watanabe et al., 2014*; *Vaden et al., 2019*; reviewed by *Rizzoli and Betz, 2005*, *Neher, 2015*; *Chamberland and Tóth, 2016*). It is still rather unclear whether functionally heterogeneous SV pools are structurally identifiable and thus support diverse forms of synaptic transmission and would also play a pivotal role in long- and short-term plasticity. Synaptic transmission can be modulated in various ways depending on the availability of SVs and on their recycling rates. Hence, the size of both the RRP and RP critically determines synaptic efficacy, strength, and plasticity. These parameters are controlled at the PreAZ but vary substantially across various CNS synapses (reviewed by *Rizzoli and Betz, 2005*; *Neher, 2015*). The contribution of the RRP size to synaptic dynamics and the mechanisms by which such control is achieved at individual SBs remains largely unknown.

L1-SBs had a total pool size of ~3500 SVs/AZ, larger than in other layers of the human TLN: ~2-fold in L4 (~1800 SVs; *Yakoubi et al., 2019b*), ~2.6-fold in L5 (~1350 SVs; *Yakoubi et al., 2019a*), and ~3-fold in L6 (~1150 SVs; *Schmuhl-Giesen et al., 2022*). Remarkably, the total pool sizes in L4 and L5 of the human TLN were significantly larger than those in L4 (~550 SVs/AZ; *Rollenhagen et al., 2015*) and L5 (~750 SVs/AZ; *Rollenhagen et al., 2018*) of the rat somatosensory cortex.

The largest total pool size in L1 may point to a pivotal role in the rapid replenishment after depletion by high-frequency stimulation and the transfer of SVs into the RRP and RP from the total pool in L1 SBs.

The putative RRP at L1 PreAZs was on average ~4 SVs/AZ for the p10 nm RRP and similar to that in L5 (~5 SVs/AZ), but significantly smaller by ~5.3-fold to that of L4 (~20 SVs/AZ) and by ~3.5-fold when compared to L6 in the human TLN. Also, huge differences were observed in the putative p20 nm RRP, which at L1 PreAZs was ~20 SVs, ~40 SVs in L4, ~15 SVs in L5, and ~30 SVs in L6, respectively. It has to be noted that a layer-specific difference in both the putative p10 nm and p20 nm RRP exists, although huge variability was observed as indicated by the SD, IQR, and variance. Hence, at L1 PreAZs, the overall RRP, taking the p10 nm and p20 nm criterion (both together are less than a vesicle diameter), was constituted by ~25 SVs rapidly available that may partially contribute to the efficacy, strength, and reliability of synaptic transmission, but also in the modulation of short-term synaptic plasticity in L1.

The size of the putative RP/PreAZ was ~470 SVs in human L1 SBs and thus ~3-fold larger than that in L6 SBs, ~1.3-fold larger than in L4, and ~2.4-fold larger than in L5 SBs of the human TLN. In the rodent neocortex, the RP/PreAZ comprised ~130 (L4; *Rollenhagen et al., 2015*) and ~200 SVs (L5; *Rollenhagen et al., 2018*). The comparably largest size of the RP in L1 when compared with values for the other cortical layers in the human TLN and to that found for L4 and L5 in rodents may also point to a rapid availability of SVs from the putative RP after the replenishment of the putative RRP during high-frequency stimulation. If the refilling rates were activity dependent, the large size of the RP at L1 PreAZs could explain some forms of short-term synaptic plasticity, for example a substantial increase in synaptic strength during frequency facilitation and post-tetanic potentiation at these SBs.

Finally, the putative resting pool of SVs at L1 PreAZs in the human TLN is also relatively large (~3000 SVs) and again the largest when compared with other layers in the human TLN (L4 and L5: ~1250 SVs; L6: ~900 SVs). The size of the resting pool may guarantee to rapidly replenish the RRP and RP after repetitive high-frequency stimulation via active transfer of SVs with the help of mitochondria associated with the pool of SVs (this study, see also *Zhou and Fuster, 1996*; *Verstreken et al., 2005*; *Smith et al., 2016*).

The notable disparities in AZ and SV pool sizes among individual SBs may contribute to rapid alterations in the computational properties of single neurons or networks. Consequently, these variations in AZ and SV pool sizes at L1 SBs may critically influence the behavior of SBs during so-called Up-and-Down states as described for other SBs of the CNS (*Zhou and Fuster, 1996*; *Sanchez-Vives and McCormick, 2000*; *Sakata and Harris, 2009*; *Testa-Silva et al., 2014*).

## Importance of presynaptic mitochondria in synaptic transmission

Mitochondria in the cortical layers of the human TLN were organized in clusters associated with the pool of SVs (this study, see also *Yakoubi et al., 2019a*, *Yakoubi et al., 2019b*; *Schmuhl-Giesen et al., 2022*). In L1, mitochondria only contribute by about 7.2% (L1a) and 6.7% (L1b) to the total SB volume with similar values between L1 and L6, but their percentage was ~2-fold lower than values in L4 and L5, suggesting a layer-specific distribution of mitochondria in the human TLN.

Mitochondria play a pivotal role in the recruitment and mobilization of SVs from the RP and resting pool and in the priming and docking process of SVs (*Verstreken et al., 2005*; *Perkins et al., 2010*; *Smith et al., 2016*; reviewed by *Dallérac et al., 2018*). In the CNS, they act as the main source of internal calcium (*Pozzan and Rizzuto, 2000*; *Rizzuto et al., 2000*), thus they regulate and control the internal calcium concentrations in CNS terminals required for the signal cascades where, for example, synaptic proteins driven by $Ca^{2+}$ like synaptotagmin, synaptophysin, synaptobrevin, and the SNARE-complex are involved (*Südhof, 2002*).

## Astrocytic coverage of L1 SBs in the human TLN

It is widely recognized that astrocytes play a crucial role in the formation of the 'tripartite' synapse, which is a common characteristic of cortical synapses. Astrocytes serve both as a physical barrier to glutamate diffusion and to mediate neurotransmitter uptake via transporters, thereby regulating the spatial and temporal concentration of neurotransmitters in the synaptic cleft (*Oliet et al., 2004*; *Min and Nevian, 2012*; *Pannasch et al., 2014*; reviewed by *Dallérac et al., 2018*). In addition, they modulate synaptic transmission by activating pre- and postsynaptic receptors (*Haydon and Carmignoto, 2006*; *Le Meur et al., 2012*). Moreover, it was found that the control of timing-dependent long-term depression (t-LTD) at neocortical synapses is critically influenced by astrocytes by increasing $Ca^{2+}$ signaling during the induction of t-LTD (*Min and Nevian, 2012*).

Remarkably, significant layer-specific differences exist in the astrocytic coverage of synaptic complexes in the human TLN. Although in L4 and L5 ~60–80% of the total volumetric fraction was occupied by astrocytic processes and most of AZs in L4 were tightly ensheathed by fine astrocytic processes, in L1 and L6 the volumetric fraction of astrocytic processes was only ~20% with a loose and incomplete coverage of AZs in L1 (*Figure 6D*). As a consequence, the lack or only incomplete astrocytic coverage of synaptic complexes in L1 can only partially act as a physical barrier to neurotransmitter diffusion. In addition, beside a vertical, also a horizontal diffusion of glutamate at the synaptic cleft is possible. Hence, two possible scenarios for the role of astrocytes are present at L1 synaptic complexes. First, as shown for only a small part (10%) of synaptic complexes in L1, the 'tripartite' synapse is realized at both sides of the synaptic cleft (*Figure 6D*). This would allow the selective uptake of horizontally diffusing glutamate via glutamate transporters located in the fine astrocytic processes. This prevents 'glutamate spillover', and thus allows vertically directed neurotransmitter diffusion and docking to postsynaptic receptors at the PSD.

The remaining larger part of synaptic complexes in L1 either lacks coverage by fine astrocytic processes at AZs or is only partially ensheathed (*Figure 6D*). Here, 'glutamate spillover' of horizontally diffusing neurotransmitter quanta is most likely, and thus synaptic crosstalk between neighboring synaptic complexes at AZs in neighboring spines on terminal tuft dendrites because spine density in L1 is high. This may cause a switch from asynchronous to synchronous release from neighboring synaptic complexes upon repetitive low- and high-frequency stimulation (*von Gersdorff and Borst, 2002*).

In summary, two distinct structural scenarios were identified regarding the role of astrocytes in L1. However, it is still rather unknown which of the two scenarios contributes more efficiently to synaptic transmission and plasticity in L1.

## The role of L1 in the information processing of the cortical column

The computational properties of the neocortex depend upon its ability to integrate the information provided by the sensory organs (bottom-up information) with internally generated signals such as expectations or attentional signals (top-down information). This integration occurs at apical tuft dendrites of L2/3 and L5 pyramidal neurons in L1. Importantly, L1 is the predominant input layer for top-down information, relayed by a rich, dense network of cortico-cortical, commissural, and associational long-range axonal projections. In experimental animals, L1 consists exclusively of a heterogeneous population of GABAergic interneurons providing feedforward and feedback inhibition (see for example *Jiang et al., 2013*; *Boldog et al., 2018*; *Obermayer et al., 2018*; *Kwon et al., 2019*; reviewed by *Huang et al., 2024*) providing signals to the terminal tuft dendrites of pyramidal neurons (reviewed by *Schuman et al., 2021*). In humans, there is a persistent subpopulation of CR-cells that provides excitation due to their long-range transcolumnar axonal projection to the terminal tuft dendrites (*Anstötz et al., 2014*), but may also control the population of GABAergic interneurons, which is as of date unknown.

Thus, L1 in the human neocortex is a central locus of neocortical associations controlled by excitation and distinct types of inhibition (reviewed by *Hartung and Letzkus, 2021*; *Schuman et al., 2021*). In summary, the population of GABAergic interneurons in L1 provides inhibition and disinhibition to pyramidal terminal tuft dendrites by either shunting or promoting the generation of $Ca^{2+}$- and $Na^+$-spikes by temporal and spatial coincidence detection. Thus, the circuitry of L1, including the persistent subpopulation of excitatory CR-cells, is an important and fundamental 'integrator' of the neocortex sampling bottom-up information within or even across cortical columns, but also acts as a 'filter' or 'discriminator' and also as an 'amplifier' in the information processing of the neocortex.

Hence, L1 of the neocortex can be regarded as a complex and intricate layer of the brain serving different functions, for example to gate inputs, convey expectations and context as well as mediate states of consciousness, attention, cross-modal interactions, sensory perception, and learning (*Bastos et al., 2012*; *Heeger, 2017*; *Zagha, 2020*; reviewed by *Gilbert and Li, 2013*). There is evidence that L1 processing, particularly $Ca^{2+}$ signals in the distal apical dendrite, is important for sensory perception (*Xu et al., 2012*).

Recent studies have revealed that L1 plays a crucial role in top-down attentional processes, aiding in directing attention, based on our goals and intentions (*Hartung and Letzkus, 2021*; *Schuman et al., 2021*). Additionally, L1 is implicated in the formation and consolidation of long-term memories,

with disruptions in L1 function associated with memory deficits. Furthermore, L1 contributes to brain plasticity, which involves the brain's ability to change and adapt in response to experience, including the process by which synapses alter in strength and number (reviewed by *Hartung and Letzkus, 2021*; *Schuman et al., 2021*).

Overall, L1 of the neocortex is a complex and fascinating layer of the brain that is involved in a wide range of cognitive processes and brain plasticity. Since this is relayed by a network of SBs in L1, their composition and connections play a pivotal role in columnar information processing.

## Materials and methods

Human brain tissue sampling during epilepsy surgery has been provided by Dr. med. Dorothea Miller, PD Dr. med. Marec von Lehe, Department of Neurosurgery, Knappschaftschafts/Universitäts-Krankenhaus Bochum and were approved by the Ethical Committees of the Rheinische Friedrich-Wilhelms-University/University Hospital Bonn (Ethical votum of the Medical Faculty to Prof. Dr. med. Johannes Schramm and Prof. Dr. rer. nat. Joachim Lübke, Nr. 146/11), the University of Bochum (Ethical votum of the Medical Faculty to PD Dr. med. Marec von Lehe and Prof. Dr. rer. nat. Joachim Lübke, Reg. No. 5190-14-15; and renewed Ethical votum of the Medical Faculty to Dr. med. Dorothea Miller and Prof. Dr. rer. nat. Joachim Lübke, Reg. No. 17–6199-BR). The consent of the patients was obtained by written and signed statements, and all further experimental procedures were approved by the same Ethical Committees cited above, and the EU directive (2015/565/EC and 2015/566/EC) concerning working with human tissue used for experimental and scientific purposes. All subsequent experimental procedures were approved by the Research Committee of the Research Centre Jülich GmbH. To meet the German protection of data privacy, the patient's identity is coded.

### Fixation and tissue processing for TEM

Tissue samples from the human TLN were, after their removal, prepared and embedded for conventional TEM and TEM tomography analysis. All neocortical access tissues were obtained from patients suffering from drug-resistant temporal lobe epilepsy (1 male and 3 female, 24–65 years in age for L1, and additionally 2 male and 3 female, 25–63 in age for L4; see also *Supplementary file 2*).

The pre-surgical work-up comprised high-resolution magnetic resonance imaging together with long-term video-electro-encephalography (EEG)-monitoring and electrophysiology using multiple electrode recordings. In all cases, the circumscribed epileptic focus was in the hippocampus proper, but not in the neocortical access regions of the TL.

During epilepsy surgery, blocks of both non-affected, non-epileptic and epileptic neocortical access tissue samples (see *Supplementary file 2*) were resected, and part of the tissue samples was further histologically examined by neuropathologists. The non-affected non-epileptic neocortical access tissue samples were always taken far from the epileptic focus and may thus be regarded as non-epileptic, as also demonstrated by functional studies using the same experimental approach (*Mohan et al., 2015*; *Molnár et al., 2016*; *Seeman et al., 2018*; *Obermayer et al., 2018*; reviewed by *Mansvelder et al., 2019*). These studies have clearly demonstrated that neocortical access tissue samples taken from epilepsy surgery do not differ in electrophysiological properties and synaptic physiology when compared with acute slice preparations in experimental animals.

So-called 'post mortem' tissue was not used in this study since the ultrastructural quality (preservation) of such material is not suitable enough for fine-scale high-resolution TEM, due to severe distortions of relevant structural features, for example fragmentation and lysis of membranes of SBs, PreAZs, PSDs, and SVs that are required for the generation of quantitative 3D-models of SBs and their target structures (Lübke, personal observation). However, under certain conditions, also 'post mortem' tissue samples can be used (*Domínguez-Álvaro et al., 2019*, 2921; *Cano et al., 2021*; *Cano et al., 2023*).

Inter-individual differences were found in the structural and synaptic parameters analyzed as shown by the box plots (Appendix 1, 2; *Source data 2*).

Other recent studies using the same experimental approach neglected and thus discarded the effect of the pharmacological treatments and disease condition (*Alonso-Nanclares et al., 2008*; *Testa-Silva et al., 2014*; *Mohan et al., 2015*; *Molnár et al., 2016*; *Seeman et al., 2018*; *Domínguez-Álvaro et al., 2019*; *Yakoubi et al., 2019a*, *Yakoubi et al., 2019b*; *Schmuhl-Giesen et al., 2022*).

For the study of L1 in the human TLN, blocks of neocortical access tissue were sampled from the temporo-lateral or temporo-basal regions of the inferior temporal gyrus and the gyrus medialis. Immediately after their removal during epilepsy surgery, biopsy samples of the TLN were immersion-fixed in ice-cold 4% paraformaldehyde and 2.5% glutaraldehyde diluted in 0.1 M phosphate buffer (PB, pH 7.4) for 24–72 hr at 4 °C. The fixative was replaced by fresh fixative after 2 hr and changed twice during the subsequent fixation period. Prior to vibratome sectioning, brain tissue samples were thoroughly rinsed in ice-cold PB and afterwards embedded in 5% Agar-Agar (Sigma, Munich, Germany) diluted in PB.

Neocortical tissue blocks were cut in the coronal plane through the TLN with a Vibratome VT 1000 S (Leica Microsystems GmbH, Wetzlar, Germany) into 150–200 µm thick sections, collected in ice-cold PB, and washed again several times in PB. Afterwards, they were transferred to 0.5–1% PB-sucrose buffered Osmium tetroxide (OsO$_4$, 300 mOsm, pH 7.4; Sigma, Munich, Germany) for 60–90 min. After visual inspection to check for the quality of post-fixation, sections were thoroughly washed several times in PB and left overnight at 4 °C in PB. The next day, they were dehydrated in an ascending series of ethanol starting at 20%, 30%, 50%, 60%, 70%, 80%, 90%, 95% to absolute ethanol (15 min for each step and absolute ethanol, 30 min twice), followed by a brief incubation in propylene oxide (2 min twice; Fluka, Neu-Ulm, Germany). Sections were then transferred into a mixture of propylene oxide and Durcupan resin (2:1, 1:1 for 1 hr each; Fluka, Neu-Ulm, Germany) and stored overnight in pure resin. The next day, sections were flat embedded on coated glass slides in fresh Durcupan, coverslipped with Acla foils and polymerized at 60 °C for 2 days.

## Semi- and ultrathin sectioning

After light microscopic (LM) inspection, a tissue block containing the region of interest (ROI) was glued on a pre-polymerized block and trimmed down. Semithin sections were cut with a Leica UltracutS ultramicrotome (Leica Microsystems, Vienna, Austria), with a Histo-Diamond knife (Fa. Diatome, Nidau, Switzerland). Afterwards they were briefly stained with methylene-blue (Sigma-Aldrich Chemie GmbH, Taufkirchen, Germany) to identify the cortical layers, particularly L1 and underlying L2, examined and documented using a motorized Olympus BX61 microscope equipped with the Olympus CellSense analysis hard- and software (Olympus GmbH, Hamburg, Germany). All images were stored in a database until further use.

After further trimming of the block to its final size, serial ultrathin sections (50±5 nm thickness; silver to silver gray interference contrast) were cut with a Leica UltracutS ultramicrotome through the determined ROI of L1a and L1b, respectively. Ultrathin sections in serial sequence were collected on Pioloform-coated slot copper grids (Plano, Wetzlar, Germany). Prior to TEM examination, sections were stained with 5% aqueous uranyl acetate or Uranyless (Science Services, Munich, Germany) for 15–20 min and lead citrate for 3–5 min according to *Reynolds, 1963* to enhance the contrast of biological membranes at the TEM level. In each series of ultrathin sections, two or three ROIs throughout L1a or L1b were chosen and photographed at a final TEM magnification of ×8000 with a Zeiss Libra 120 (Fa. Zeiss, Oberkochen, Germany) equipped with a Proscan 2 K digital camera (Fa. Tröndle, Moorenweis, Germany) using the ImageSP software and its panorama function (Fa. Tröndle, Moorenweis, Germany). In addition, interesting details of synaptic structures in L1a and L1b were taken at various TEM magnifications.

All images were stored in a database until further use. Selected TEM images for publication were further edited using Adobe Photoshop and Adobe Illustrator software.

## 3D-volume reconstructions and quantitative analysis of L1 SBs

TEM panorama images composing each series were imported, stacked, and aligned in the reconstruction software OpenCAR (Contour Alignment Reconstruction; for details see *Sätzler et al., 2002*). The main goal of this study was to quantify several structural and synaptic parameters representing structural correlates of synaptic transmission and plasticity. Excitatory SBs were characterized by large round SVs and prominent PreAZs and PSDs in contrast to putative GABAergic terminals that have smaller, more oval-shaped SVs and thin or no PSDs.

The following structural parameters were analyzed in randomly selected excitatory SBs: (1) surface area and volume of SBs; (2) volume of mitochondria; (3) surface area of PreAZs (*Dufour et al., 2016*) and PSDs; two apposed membrane specializations separated by the synaptic cleft; (4) number and

diameter of clear SVs and dense-core vesicles (DCVs); and (5) distance of individual SVs from the PreAZ for the structural definition of the RRP, RP, and resting pool. 3D-volume reconstructions were generated by drawing contour lines on the structures of interest in each panorama image within a series of images until a given structure was completed. Presynaptic SBs, their mitochondria as well as their postsynaptic target structures were outlined on the outer edge of their membranes, using closed contour lines throughout the entire stack within a series. A SB was considered completely captured when it was possible to follow the axon in both directions through the entire series (*en passant* SBs) or the enlargement of the axon leading to an end-terminal SB. The beginning of a synaptic terminal was defined by the typical widening of the axon and the abrupt occurrence of a pool of SVs.

The PreAZs and PSDs were regarded as complete when their perimeters were entirely reconstructed in a series of ultrathin sections. The surface areas of the PreAZ and PSD were computed separately by first generating a 3D surface model of the SB. The PreAZ was then measured by extracting this area from the reconstructed presynaptic bouton membrane that was covered by this membrane specialization. Hence, the length (L) of the PreAZ (L PreAZ) and the surface area (SA) of the PreAZ (SA PreAZ) is already known.

The size of the PSD opposing the PreAZ was estimated under the following assumptions: (1) both membrane specializations, PreAZ and PSD, run parallel to each other at the pre- and postsynaptic apposition zone; (2) for both membrane specializations a contour line was drawn determining their actual length (L PreAZ and L PSD). Hence, the surface area of the PSD (SA PSD) is estimated by the following equation:

$$SA\text{PSD} = SA\text{PreAZ} \times L\text{PSD}/L\text{PreAZ}$$

which is the perimeter ratio between the outlines of the PSD to that of the synaptic contact.

The synaptic cleft width was measured because of its importance for the transient temporal and spatial increase of the glutamate concentration, reversible binding of glutamate to appropriate glutamate receptors, and eventual uptake and diffusion of glutamate out of the synaptic cleft. To a large extent, these processes are governed by the geometry of this structure and the shape and size of the PreAZs and PSDs.

All SVs were marked throughout each SB, their diameters were individually measured, and their distances to the PreAZ were automatically detected using an algorithm implemented in OpenCAR (minimal distance between each SV membrane to the contour line of the PreAZ) throughout the SB in every single image of the series. Large DCVs were only counted in the image where they appeared largest. To avoid double counts, only clear ring-like structures were counted as SVs. However, SVs might be missed in densely packed regions, because ring-like traces may partly overlap. This effect may counteract any double counts. Based on the small extent and the partially counteracting nature of this effect, the numbers of small clear vesicles reported in this study remained uncorrected (*Yakoubi et al., 2019b*, *Schmuhl-Giesen et al., 2022*).

In this work, aldehyde fixation was used that is thought to induce tissue shrinkage, thereby biasing structural quantification (*Eyre et al., 2007*; *Korogod et al., 2015*). A direct comparison of structural parameters obtained from either aldehyde or cryo-fixed and substituted tissue samples (*Korogod et al., 2015*) showed differences in cortical thickness (~16% larger in cryo-fixed material), in the volume of extracellular space (~6-fold larger in cryo-fixed material), a slight increase in glial volume and overall density of synaptic contacts (~14% in cryo-fixed material), but no significant differences in neuronal structures such as axons, dendrites, and SV diameter.

In the structural and synaptic parameters as estimated here, no significant difference was found for SB size and other synaptic subelements such as mitochondria, active zones (AZs), and SVs when compared with other studies (*Zhao et al., 2012a*, *Zhao et al., 2012b*). Therefore, no corrections for shrinkage were applied, and we are thus convinced that the synaptic parameters reported here are accurate and can be directly used for detailed computational models. In addition, large-scale preservation for ultrastructural analysis will therefore continue to rely on chemical fixation approaches, due to the limited preservation of the ultrastructure in cryo-fixed material as stated in *Korogod et al., 2015*.

## Golgi-Cox impregnation of biopsy material in L1 of the human TLN

To get an impression about the neuronal organization of the human TLN, four tissue blocks were processed with the Golgi-Cox impregnation technique using the commercially available Hito Golgi-Cox OptimStain kit (Hitobiotec Corp, Kingsport, TE, USA). After removal of the biopsy samples, tissues were briefly rinsed twice in double distilled water (dd H$_2$O), and then transferred into impregnation solution 1 overnight at room temperature. The next day, tissue samples were incubated in fresh impregnation solution 1 and stored for 14 days in the dark at room temperature. The sample blocks were then transferred in solution 3 and kept in the dark at room temperature for one day. Thereafter, they were placed into fresh solution 3 in the dark at room temperature for 6 additional days. Then, solution 3 was exchanged, and samples were stored at 4 °C in the dark overnight. Tissue blocks were embedded in 5% agarose (Carl Roth, Karlsruhe, Germany) diluted in ddH$_2$O and sectioned with a vibratome in the coronal plane at 100–250 μm thickness and then transferred to ddH$_2$O.

After careful removal of the agarose, free-floating sections were incubated into solution 3 for 2–3 min in the dark at room temperature, and right after placed into ddH$_2$O, washed several times, and stored overnight. Afterwards, they were rinsed twice in ddH$_2$O for 4 min each, and dehydrated in 50%, 70%, and 95% ethanol for 5 min each, then transferred into absolute ethanol (3×5 min), defatted in xylene, embedded in Eukitt (Sigma-Aldrich Chemie GmbH, Taufkirchen, Germany), finally coverslipped and air-dried. Afterwards, sections were examined and imaged with an Olympus BX 61 LM equipped with the CellSense software package (Olympus, Hamburg, Germany) at various magnifications, and images were stored in a database until further use.

## Stereological estimation of the density of L1 synaptic contacts in the human TLN

The density of synaptic complexes, composed either between an SB with a dendrite or spine, in a given volume is a valuable parameter to assess the structural and functional changes in the brain, which are linked to age, pathological or experimental conditions (*Rakic et al., 1994*; *DeFelipe et al., 1999*). The density of synaptic contacts was unbiasedly estimated in L1, separated for L1a and L1b, from four patients, respectively (*Supplementary file 2*; *Source data 1*) using the physical dissector technique (*Mayhew, 1996*; *Fiala and Harris, 2001*) by counting the synaptic complexes in a virtual volume generated by two adjacent ultrathin sections that is the dissector: the reference section and the look-up section. Here, counting was performed using FIJI (*Schindelin et al., 2012*) on a stack of 20 aligned serial electron micrographs for each patient taken from the series of ultrathin sections used for the 3D-volume reconstructions of SBs in L1. An unbiased counting frame was first set, and synaptic contacts to be considered (counted) are the ones present in the reference section only and meeting the following criteria: presence of a PreAZ and a prominent or thin PSD separated by a synaptic cleft and SVs in the presynaptic terminal. Care was taken to distinguish between excitatory and inhibitory synaptic contacts, as well as the postsynaptic target structures (dendritic shafts or spines). Finally, the density of synaptic contacts (*Nv*) per 1 mm$^3$ was calculated using the formula below:

$$Nv = \sum Qd / \sum Vd$$

where *Qd* is the number of synaptic contacts per dissector and Vd is the volume of the dissector given by: Number of dissectors x frame area x section thickness.

## TEM tomography of L1 SBs in the human TLN

TEM tomography was performed on 200–300 nm thick sections cut from blocks prepared for serial ultrathin sectioning as described above (*Table 2*; *Source data 3*). Sections were mounted on pioloform-coated line copper grids and were counterstained with uranyl acetate and lead citrate following a slightly modified staining protocol as described by *Reynolds, 1963*.

Subsequently, sections were examined with a JEOL JEM 1400Plus, operating at 120 kV and equipped with a 4096×4,096 pixels CMOS camera (TemCam-F416, TVIPS, Gauting, Germany). Tilt series were acquired automatically over an angular range of −60° to +60° at 1°-degree increments using Serial TEM (*Mastronarde, 2005*). Stack alignment and reconstruction by filtered backprojection were carried out using the software package iMOD (*Kremer et al., 1996*). Final reconstructions were ultimately filtered using a median filter with a window size of 3 pixels. Tilt series were stored as .tif

files and were further processed using the freely available software Imod 4.9.12 and ImageJ (ImageJ, RRID:SCR-0033070). In each tilt series, so-called 'docked' SVs identified by their fusion with the PreAZ or as omega-shaped bodies that had already released a quantum of neurotransmitter were counted separately for L1a and L1b and for their target structures, dendritic shafts or spines in randomly selected SBs.

To determine possible differences to the values obtained by the 3D reconstructions, in the same tilt series binned at 24 nm (section thickness), the diameter of SVs was measured randomly by using OpenCAR as described above. To avoid double counts and underestimations of the diameter, the ring-shaped structure of an SV was compared on each tilted slice and only marked if it appeared largest. In 18 tilt series, each consisting of 9–14 tilt slices, randomly selected SBs were investigated (L1a: 8 tilt series; 16 SBs with a total of 2094 SVs; L1b: 10 tilt series; 22 SBs with a total of 3740 SVs).

## Quantitative analysis of the astrocytic coverage

Although methodologically associated with the possibility of a certain degree of error, astrocytic processes around synaptic complexes were mainly identified by their relatively clear cytoplasm compared to synaptic boutons or dendritic profiles; in addition by their irregular, stellate shape, by the presence of glycogen granules and bundles of intermediate filaments (*Peters et al., 1991*; *Ventura and Harris, 1999*). With greater accuracy, astrocytic processes can be identified with immunolabeling, but this leads to poorer tissue preservation and resolution of cellular structures such as AZs. Among studies in rats (e.g. *Rollenhagen et al., 2015*; *Rollenhagen et al., 2018*) in a previous study on the human TLN (*Yakoubi et al., 2019b*), immunohistochemistry against glutamine synthetase, a key enzyme in astrocytes, was carried out. Since similar results were obtained without immunohistochemistry when identifying astrocytic processes around AZs, immunohistochemistry was omitted in the series of follow-up studies on the human TLN.

To quantify the volume contribution of astrocytic processes in the human neocortex, the interactive software ImageJ (*Schneider et al., 2012*) was used. The first, the middle, and the last images of an individual TEM series were used for a further quantitative volumetric analysis (*Source data 4*, sheet 01). In each section of the same series used for the 3D-volume reconstructions, a grid (grid size 1×1 μm$^2$) was placed over the TEM image, and in each square, the abundance of fine astrocytic processes was documented throughout these images and averaged. Using the Cavalieri method [Unbiased Stereology: Three-Dimensional Measurement in Microscopy (Advanced Methods) Paperback-January 7, 2005, by Vyvyan Howard Matthew Reed], the (absolute) volume contribution of astrocytic processes was determined according to the Cavalieri estimator:

$$V = a(p) \times \sum P \times t$$

where a(p) is the size of one square (0.8×0.8 μm2), P is the number of squares counted, and t is the thickness of the slice.

In addition, in consecutive sections spanning the complete AZ, the astrocytic coverage was evaluated under the following criteria: fine astrocytic processes were observed on (1) both sides of the AZ reaching as far as the synaptic cleft, (2) only one side of the AZ was covered by fine astrocytic processes and (3) AZs without astrocytic processes (*Source data 4*, sheet 02). Values obtained for L1 (L1a: n=3 patients; L1b: 3 patients) either for SBs terminating on dendritic spines (L1a: n=128; L1b: n=126) or shafts (L1a: n=48; L1b: n=44) were compared to L4 (n=5 patients; dendritic spines n=93; dendritic shafts n=41). Layer 4 was chosen because the volumetric analysis showed the highest portion of astrocytic processes in this layer compared to all other cortical layers. All values are given as percentages.

## Statistical analysis

The mean value ± standard deviation (SD), the median with the interquartile range (IQR), the coefficient of variation (CV), skewness, variance, and the coefficient of correlation (R$^2$) were given for each structural parameter analyzed. The p-value was considered significant only if p<0.05. Box and Violin plots (Plotly 4.0.0 https://chart-studio.plotly.com) were generated to investigate inter-individual differences for each patient and structural parameter (Appendix 1, 2; *Source data 2*).

To test for significant differences, the non-parametric Kruskal-Wallis H-test with a subsequent Mann-Whitney U-test analysis was performed, using PAST 4.02 (*Hammer et al., 2001*). Correlation graphs

between several structural parameters were then generated (*Figure 4*). The $R^2$-values were interpreted as follows: 0, no linear correlation; 0–0.5, weak linear correlation; 0.5–0.8, good linear correlation; and 0.8–1.0, strong linear correlation. Furthermore, a freely available Fisher's *r*-to-*z*-transformation calculator (Fisher's z) was used to test for differences in $R^2$ between L1a and L1b (p-value <0.05). For statistical comparison with other layers of the human TLN, L1a and L1b were grouped together as L1.

Parts of the data acquisition were carried out in the context of a PhD Thesis (*Sadeghi, 2024*). Values obtained in the present study for L1 were compared partially to those obtained for L2, L3 (Girwert Ch., PhD Thesis in preparation), L4, L5 (*Yakoubi et al., 2019a*, *Yakoubi et al., 2019b*; Yakoubi R., PhD Thesis 2023), and L6 (*Schmuhl-Giesen et al., 2022*).

## Acknowledgements

We would like to thank our technicians Brigitte Marshallsay and Tayfun Palaz for their excellent technical assistance. Furthermore we are grateful to Dr. Dorothea Miller (Department of Neurosurgery, Knappschaftskrankenhaus; Bochum, Germany) and Prof. Marec von Lehe (now: Medizinische Hochschule Brandenburg and Universitätsklinikum Ruppin Brandenburg Universitätsklinikum der Medizinischen Hochschule Brandenburg Klinik für Neurochirurgie und Wirbelsäulenchirurgie; Neuruppin, Germany) by providing us with human donor tissue samples. Finally, the constant financial support of the Helmholtz Society and the Research Centre Jülich GmbH is very much acknowledged.

## Additional information

### Competing interests

Astrid Rollenhagen: is affiliated with Jülich GmbH.The author has no other competing interests to declare. Akram Sadeghi, Joachim HR Lübke: is affiliated with Jülich GmbH. The author has no other competing interests to declare. The other authors declare that no competing interests exist.

### Funding

| Funder | Grant reference number | Author |
| --- | --- | --- |
| Helmholtz Association | | Joachim HR Lübke |

The funders had no role in study design, data collection and interpretation, or the decision to submit the work for publication.

### Author contributions

Astrid Rollenhagen, Conceptualization, Data curation, Supervision, Investigation, Methodology, Project administration, Writing – review and editing; Akram Sadeghi, Investigation, Methodology, Writing – original draft; Bernd Walkenfort, Data curation, Formal analysis, Investigation, Visualization, Methodology; Claus C Hilgetag, Data curation, Formal analysis, Validation, Investigation, Visualization, Methodology, Writing – original draft; Kurt Sätzler, Data curation, Software, Formal analysis, Validation, Investigation, Visualization, Methodology; Joachim HR Lübke, Conceptualization, Data curation, Formal analysis, Supervision, Funding acquisition, Validation, Investigation, Visualization, Methodology, Writing – original draft, Project administration, Writing – review and editing

### Author ORCIDs

Astrid Rollenhagen ⬤ https://orcid.org/0000-0003-4887-9272
Bernd Walkenfort ⬤ https://orcid.org/0000-0002-7789-3340
Joachim HR Lübke ⬤ https://orcid.org/0000-0002-4086-3199

### Ethics

Human brain tissue sampling during epilepsy surgery have been provided by Dr. med. Dorothea Miller, PD Dr. med. Marec von Lehe, Department of Neurosurgery, Knappschaftschafts/Universitäts-Krankenhaus Bochum and were approved by the Ethical Committees of the Rheinische Friedrich-Wilhelms-University/University Hospital Bonn (Ethical votum of the Medical Faculty to Prof. Dr. med. Johannes Schramm and Prof. Dr. rer. nat. Joachim Lübke, Nr. 146/11), the University of Bochum

(Ethical votum of the Medical Faculty to PD Dr. med. Marec von Lehe and Prof. Dr. rer. nat. Joachim Lübke, Reg. No. 5190-14-15; and renewed Ethical votum of the Medical Faculty to Dr. med. Dorothea Miller and Prof. Dr. rer. nat. Joachim Lübke, Reg. No. 17–6199-BR). The consent of the patients was obtained by written and signed statements, and all further experimental procedures were approved by the same Ethical Committees cited above, and the EU directive (2015/565/EC and 2015/566/EC) concerning working with human tissue used for experimental and scientific purposes. All subsequent experimental procedures were approved by the Research Committee of the Research Centre Jülich GmbH. To meet the German protection of data privacy the patient's identity is coded.

Reviewer #2: https://doi.org/10.7554/eLife.99473.5.sa1
Reviewer #1: https://doi.org/10.7554/eLife.99473.5.sa2
Author response https://doi.org/10.7554/eLife.99473.5.sa3

## Additional files

### Supplementary files
MDAR checklist

Source data 1. Measurements of the synaptic density in L1a and L1b of the human TLN.

Source data 2. Analysis of structural and functional parameters of SBs in L1a and L1b of the human TLN.

Source data 3. Measurements of 'docked' SVs (DV) in SBs obtained by EM-Tomography in L1a and L1b of the human TLN.

Source data 4. Quantitative analysis of the astrocytic coverage of AZs in L1 and L4 of the human TLN.

Supplementary file 1. Comparison of the synaptic density between L1a, L1b, and L4 – L6 of the human TLN.

Supplementary file 2. Patient's identity and medical background.

### Data availability
All data generated or analysed during this study are included in the manuscript and supporting files; source data files have been provided.

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

## Appendix 1

### Box plots of various structural parameters in L1 of the human TLN

Data distributions for each patient in L1a (left side of dashed line) and L1b (right side of dashed line) are indicated by the medians (horizontal bars), IQRs (framed areas), minimum and maximum (vertical lines) for the distribution of: A, Surface area of SBs; B, Volume of SBs; C, Surface area of PreAZs; D, Surface area of PSD; E, Volume of mitochondria. Significant differences between different patients are indicated by asterisks. Note that several structural parameters are significantly different. The significant bars highlighted in red indicate significant differences between sublaminae L1a and L1b in the same patient. *: p<0.05; **: p<0.01; ***: p<0.001 (*Source data 2*). Hu_01 and Hu_02 were selected for both L1a and L1b in separate serial sections each.

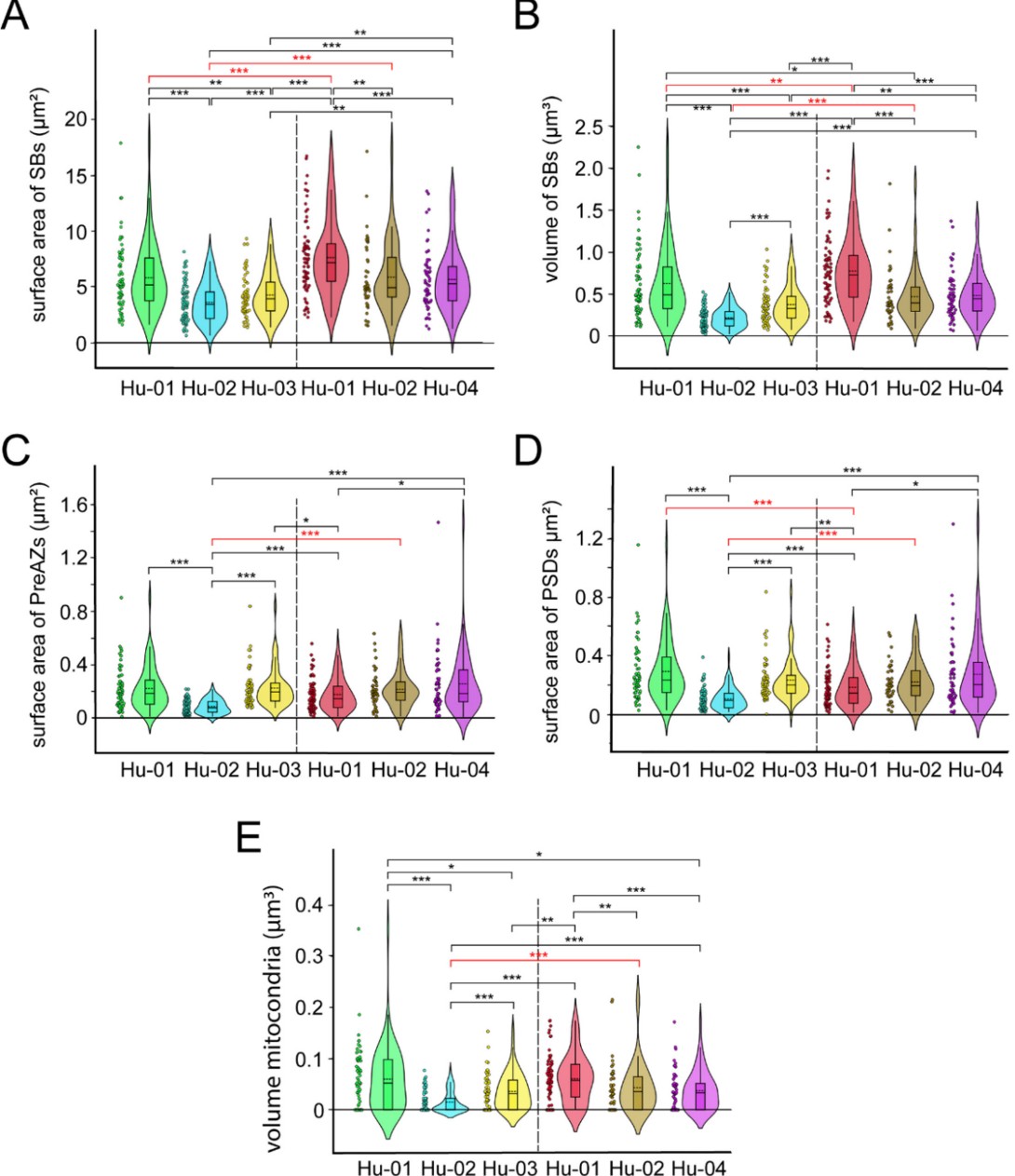

**Appendix 1—figure 1.** Box plots of various structural parameters in L1 of the human TLN. Data distributions for each patient in L1a (left side of dashed line) and L1b (right side of dashed line) are indicated by the medians (horizontal bars), IQRs (framed areas), minimum and maximum (vertical lines) for the distribution of: A, Surface area

*Appendix 1—figure 1 continued on next page*

*Appendix 1—figure 1 continued*

of SBs; B, Volume of SBs; C, Surface area of PreAZs; D, Surface area of PSD; E, Volume of mitochondria. Significant differences between different patients are indicated by asterisks. Note that several structural parameters are significantly different. The significant bars highlighted in red indicate significant differences between sublaminae L1a and L1b in the same patient. *: p < 0.05; **: p < 0.01; ***: p < 0.001 (*Source data 2*). Hu_01 and Hu_02 were selected for both L1a and L1b in separate serial sections each.

## Appendix 2

### Box plots of various synaptic parameters in L1 of the human TLN

Data distributions for each patient in L1a (left side of dashed line) and L1b (right side of dashed line) are indicated by the medians (horizontal bars), IQRs (framed areas), minimum and maximum (vertical lines) for the distribution of: **A,** Total pool of SVs; **B,** SVs in the p10 nm RRP; **C**, SVs in the p20 nm RRP; **D**, SVs in the p6-200 nm RP; **E**, SVs in the >p200 nm resting pool; Note that several structural parameters are not significantly different. The significant bars highlighted in red indicate significant differences between sublaminae L1a and L1b in the same patient. *: $p<0.05$; **: $p<0.01$; ***: $p<0.001$ (*Source data 2*). Hu_01 and Hu_02 were selected for both L1a and L1b in separate serial sections each.

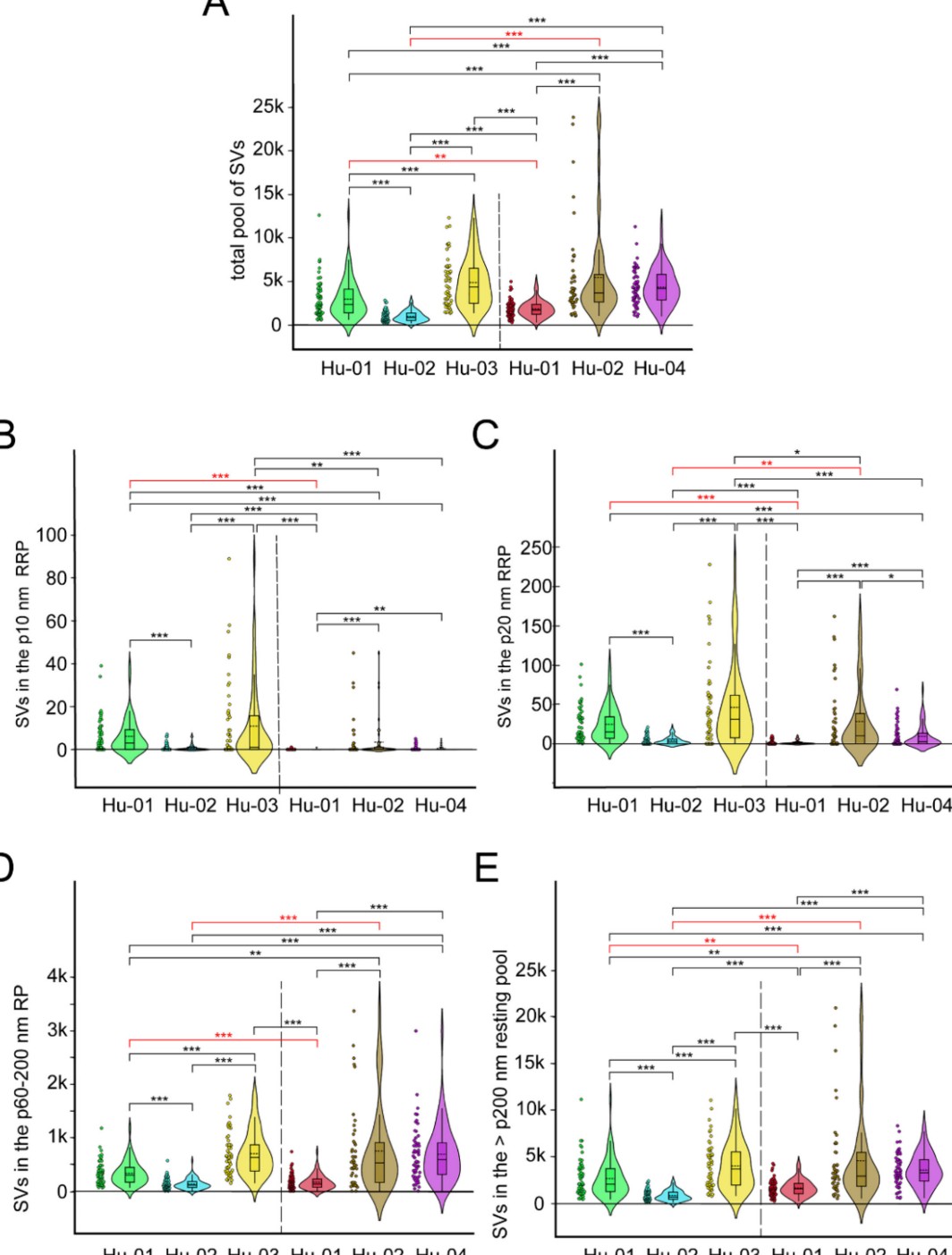

**Appendix 2—figure 1.** Box plots of various synaptic parameters in L1 of the human TLN. Data distributions for each patient in L1a (left side of dashed line) and L1b (right side of dashed line) are indicated by the medians (horizontal bars), IQRs (framed areas), minimum and maximum (vertical lines) for the distribution of: A, Total pool of SVs; B, SVs in the p10 nm RRP; C, SVs in the p20 nm RRP; D, SVs in the p6-200 nm RP; E, SVs in the > p200 nm resting pool; Note that several structural parameters are not significantly different. The significant bars highlighted in red indicate significant differences between sublaminae L1a and L1b in the same patient. *: p < 0.05; **: p < 0.01; ***: p < 0.001 (*Source data 2*). Hu_01 and Hu_02 were selected for both L1a and L1b in separate serial sections each.

