## [Editor Report · eLife Assessment]

This study provides **important** information on the ultrastructural organization of layer 1 of the human neocortex. The quantitative assessment of various synaptic parameters, astrocytic coverage and mitochondrial morphology is based on **convincing** experimental approaches. These data provide new information on the detailed morphology of human neocortical tissue that will be of interest to neuroscientists working on different network functions.

---

## [Referee Report · Reviewer #1]

Summary:

The authors investigated the anatomical features of the excitatory synaptic boutons in layer 1 of the human temporal neocortex. They examined the size of the synapse, the macular or the perforated appearance and the size of the synaptic active zone, the number and volume of the mitochondria, the number of the synaptic and the dense core vesicles, also differentiating between the readily releasable, the recycling and the resting pool of synaptic vesicles. The coverage of the synapse by astrocytic processes was also assessed, and all the above parameters were compared to other layers of the human temporal neocortex. The Authors conclude that the subcellular morphology of the layer 1 synapses is suitable for the functions of the neocortical layer, i.e. the synaptic integration within the cortical column. The low glial coverage of the synapses might allow the glutamate spillover from the synapses enhancing synaptic crosstalk within this cortical layer.

Strengths:

The strengths of this paper are the abundant and very precious data about the fine structure of the human neocortical layer 1. Quantitative electron microscopy data (especially that derived from the human brain) are very valuable, since this is a highly time- and energy consuming work. The techniques used to obtain the data, as well as the analyses and the statistics performed by the Authors are all solid, strengthen this manuscript, and support the conclusions drawn in the discussion.

---

## [Referee Report · Reviewer #2]

The study of Rollenhagen et al examines the ultrastructural features of Layer 1 of human temporal cortex. The tissue was derived from drug-resistant epileptic patients undergoing surgery, and was selected as further from the epilepsy focus, and as such considered to be non-epileptic. The analyses has included 4 patients with different age, sex, medication and onset of epilepsy. The manuscript is a follow-on study with 3 previous publications from the same authors on different layers of the temporal cortex:

Layer 4 - Yakoubi et al 2019 eLife

Layer 5 - Yakoubi et al 2019 Cerebral Cortex,

Layer 6 - Schmuhl-Giesen et al 2022 Cerebral Cortex

They find, the L1 synaptic boutons mainly have single active zone a very large pool of synaptic vesicles and are mostly devoid of astrocytic coverage.

Strengths:

The MS is well written easy to read. Result section gives a detailed set of figures showing many morphological parameters of synaptic boutons and surrounding glial elements. The authors provide comparative data of all the layers examined by them so far in the Discussion. Given that anatomical data in human brain are still very limited, the current MS has substantial relevance. The work appears to be generally well done, the EM and EM tomography images are of very good quality. The analyses is clear and precise.

Weaknesses:

The authors made all the corrections required and answered all of my concerns, included additional data sets, and clarified statements where needed.

---

## [Author Response]

The following is the authors’ response to the previous reviews.

**Public Reviews:**

**Reviewer #1 (Public review):**
Summary:The Authors investigated the anatomical features of the excitatory synaptic boutons in layer 1 of the human temporal neocortex. They examined the size of the synapse, the macular or the perforated appearance and the size of the synaptic active zone, the number and volume of the mitochondria, the number of the synaptic and the dense core vesicles, also differentiating between the readily releasable, the recycling and the resting pool of synaptic vesicles. The coverage of the synapse by astrocytic processes was also assessed, and all the above parameters were compared to other layers of the human temporal neocortex. The Authors conclude that the subcellular morphology of the layer 1 synapses is suitable for the functions of the neocortical layer, i.e. the synaptic integration within the cortical column. The low glial coverage of the synapses might allow the glutamate spillover from the synapses enhancing synaptic crosstalk within this cortical layer.Strengths:The strengths of this paper are the abundant and very precious data about the fine structure of the human neocortical layer 1. Quantitative electron microscopy data (especially that derived from the human brain) are very valuable, since this is a highly time- and energy consuming work. The techniques used to obtain the data, as well as the analyses and the statistics performed by the Authors are all solid, strengthen this manuscript, and support the conclusions drawn in the discussion.Comments on latest version:The third version of this paper has been substantially improved. The English is significantly better, there are only few paragraphs and sentences which are hard to understand (see my comments and suggestions below). Almost all of my suggestions were incorporated.

We would like to thank the reviewer for the comments and incorporated the suggestions within the latest version of the manuscript.

Remaining minor concerns:About epileptic and non-epileptic (non-affected) tissue. I am aware that temporal lobe neocortical tissue derived from epileptic patients is regarded as non-affected by many groups, and they are quite similar to the cortex of non-epileptic (tumour) patients in their electrophysiological properties and synaptic physiology. But please, note, that one paper you cited did not use samples from epileptic patients, but only tissue from non-epileptic tumor patients (Molnár et al. PLOS 2008).When you look deeper, and make thorough comparison of tissues derived from epileptic and non-epileptic patients, there are differences in the fine structure, as well as in several electrophysiological features. See for example Tóth et al., J Physiol, 2018, where higher density of excitatory synapses were found in L2 of neocortical samples derived from epileptic patients compared to non-epileptic (tumor) patients. Furthermore, the appearance of population bursts is similar, but their occurrence is more frequent and their amplitude is higher in tissue from epileptic compared to non-epileptic patients. So, I still cannot agree, that temporal neocortex of epileptic patients with the seizure focus in the hippocampus would be non-affected. Therefore I suggested to use the term biopsy tissue.

We are thankful for this comment on using non-epileptic tissue also by others. We are also aware that Molnár et al. 2008 worked with tumor tissue.

It is still not emphasized in the first paragraph of the Discussion, that only excitatory axon terminals were investigated.

We now mentioned in the first paragraph of the discussion that only excitatory synaptic boutons were investigated.

The text in the Results and the Discussion are somewhat inconsistent.The last two paragraphs of the Results section ends with several sentences which should be part of the discussion, such as line 328: This finding strongly supports multivesicular release... or line 344: --- pointing towards a layer-specific regulation of the putative RRP. Moreover, the results suggest that... and line 370: ... it is most likely... Please, correct this.

We disagree with the reviewer on these points because these sentences summarizes the findings.

The first paragraph of the Discussion summarizes the work of the quantitative EM work and gives one conclusion about the astrocytic coverage. This last sentence is inconsistent with the other parts of the paragraph. I would either write that "astrocytic coverage was also investigated" (or something similar), or move this sentence to the paragraph which discusses the astrocytic coverage.Results line 180-183. "Special connections" between astrocytic processes and synaptic boutons are mentioned, but not shown. Either show these (but then prove with staining!), or leave out this paragraph.

We deleted this paragraph as suggested.

**Reviewer #2 (Public review):**
Summary:The study of Rollenhagen et al examines the ultrastructural features of Layer 1 of human temporal cortex. The tissue was derived from drug-resistant epileptic patients undergoing surgery, and was selected as further from the epilepsy focus, and as such considered to be non-epileptic. The analyses has included 4 patients with different age, sex, medication and onset of epilepsy. The manuscript is a follow-on study with 3 previous publications from the same authors on different layers of the temporal cortex:Layer 4 - Yakoubi et al 2019 eLifeLayer 5 - Yakoubi et al 2019 Cerebral Cortex,Layer 6 - Schmuhl-Giesen et al 2022 Cerebral CortexThey find, the L1 synaptic boutons mainly have single active zone a very large pool of synaptic vesicles and are mostly devoid of astrocytic coverage.Strengths:The MS is well written easy to read. Result section gives a detailed set of figures showing many morphological parameters of synaptic boutons and surrounding glial elements. The authors provide comparative data of all the layers examined by them so far in the Discussion. Given that anatomical data in human brain are still very limited, the current MS has substantial relevance.The work appears to be generally well done, the EM and EM tomography images are of very good quality. The analyses is clear and precise.Weaknesses:The authors made all the corrections required and answered all of my concerns, included additional data sets, and clarified statements where needed.
**Recommendations for the authors:**

**Reviewer #1 (Recommendations for the authors):**
Minor suggestions:Synaptic density, lines 189-193. If you say "comparatively" high, then compare to something (cite your own work for the other layers, and tell the approximative values for the other layers). Same in line 194 comparably high to what? Other option: say "relatively high".

We corrected the sentences as suggested by the reviewer.

Line 206: When present, mitochondria (comma missing)

Corrected as suggested by the reviewer.

Line 265: Dot is missing at the end of the sentence (after Shapira et al. 2003)

Corrected as suggested by the reviewer.

Lines 300-301: Check the English for this sentence: significant difference BETWEEN TWO sublaminae and not significant difference for both sublaminae.

Corrected as suggested by the reviewer.

Lines 304-305: Check the sentence, please, it is not understandable without the text in parenthesis.

Corrected as suggested by the reviewer.

Line 354 Dot missing at the end of the sentence (after Figure 6A, B)

Corrected as suggested by the reviewer.

Line 354-358: Please rephrase this sentence (too complicated, not understandable). I do not understand why results of the L4, L5, L6 are described here. What does it mean "Astrocytes and their fine processes formed a relatively dense, but a comparably loose network within the neuropil in L1"? Dense or loose?

In the experiment measuring the volume fraction of astrocytic processes (Figure 6C), all six cortical layers were analyzed, thus we compared the values obtained for L1 with the results for L4, L5 and L6. For more clarity, we rephrased the sentence: “Astrocytes and their fine processes formed a relatively dense network in L4 and L5, but a comparably loose one within the neuropil in L1…” We also rephrased other sentences in this paragraph (as also suggested below).

Lines 359-369: Please rephrase this paragraph. The sentences are too complicated, have too many parentheses, and are not understandable. I suggest to write first how many synapses were examined in L1 and L4, then how many of them were on spine and on dendrites (either n or %). Then give the values how many (n or %) of them were "tripartite synapses", out of spine synapses and of dendritic synapses in both layers. How many of them were partially covered in both layers. Please, write the data in a systematic way. The best would be to give the values in a table as well. This way it will be more understandable (now, it is chaotic, hard to follow).

We rephrased the paragraph and added a new table (3).

Line 383: Dot missing from the end of the sentence.

Corrected as suggested by the reviewer.

Line 436: Reconsider "comparably low compared to". The comparably means what in this case? The whole paragraph is hard to understand, please, check and review for improvements to the use of English or use chatGPT to check it.

We corrected the sentence according to the reviewer’s suggestion.

Line 487: Same thing again: "The comparably largest size of the RP in L1 when compared..."What would you like to say with "comparably"? Check the meaning of this word in a dictionary, please. I have the feeling that you are using this word instead of "relatively".

Corrected as suggested by the reviewer.

Line 488 "and TO that found fot L4 and L5 in rodents..."

Corrected as suggested by the reviewer.

Line 493-495: Same again, comparably when compared, correct, please.

Corrected as suggested by the reviewer.

Supplemental figures: Now I do understand why Hu-01 and Hu-02 are twice, and I think, 3 patients were examined for L1a and three for L1b. But which side is which on the subfigures? Left side (Hu-01, 02 03) was used for L1a, or L1b? Could you write this in the legend, or mark on the figure (at least at one subfigure), please?

We implemented a comment for clarity.